# The ATM-E6AP-MASTL axis mediates DNA damage checkpoint recovery

Yanqiu Li[1], Feifei Wang[1], Xin Li[1], Ling Wang[1], Zheng Yang[2], Zhongsheng You[2], Aimin Peng[1]*

[1]Department of Oral Biology, University of Nebraska Medical Center, Lincoln, United States; [2]Department of Cell Biology and Physiology, School of Medicine, Washington University in St. Louis, St. Louis, United States

**Abstract** Checkpoint activation after DNA damage causes a transient cell cycle arrest by suppressing cyclin-dependent kinases (CDKs). However, it remains largely elusive how cell cycle recovery is initiated after DNA damage. In this study, we discovered the upregulated protein level of MASTL kinase hours after DNA damage. MASTL promotes cell cycle progression by preventing PP2A/B55-catalyzed dephosphorylation of CDK substrates. DNA damage-induced MASTL upregulation was caused by decreased protein degradation, and was unique among mitotic kinases. We identified E6AP as the E3 ubiquitin ligase that mediated MASTL degradation. MASTL degradation was inhibited upon DNA damage as a result of the dissociation of E6AP from MASTL. E6AP depletion reduced DNA damage signaling, and promoted cell cycle recovery from the DNA damage checkpoint, in a MASTL-dependent manner. Furthermore, we found that E6AP was phosphorylated at Ser-218 by ATM after DNA damage and that this phosphorylation was required for its dissociation from MASTL, the stabilization of MASTL, and the timely recovery of cell cycle progression. Together, our data revealed that ATM/ATR-dependent signaling, while activating the DNA damage checkpoint, also initiates cell cycle recovery from the arrest. Consequently, this results in a timer-like mechanism that ensures the transient nature of the DNA damage checkpoint.

*For correspondence:
aimin.peng@unmc.edu

Competing interest: The authors declare that no competing interests exist.

## eLife assessment

This study reports the **important** finding that there appears to be a timer that monitors the repair of DNA after damage and regulates whether cells are subsequently able to enter mitosis. The authors identify proteins important for this decision and propose a mechanism supported by **solid** but not conclusive data. This study will be of interest to researchers in the fields of DNA damage repair and cell cycle control.

## Introduction

DNA damage activates a wide range of cellular responses, including DNA repair, cell cycle checkpoints, and cell death. Collectively defined as the DNA damage response (DDR), these mechanisms ensure genomic integrity and prevent progression of cancer, aging, and other diseases (*Ciccia and Elledge, 2010*; *Jackson and Bartek, 2009*; *Lou and Chen, 2005*). Among these DDR pathways, the DNA damage checkpoint temporally halts the progression of the cell cycle, to facilitate DNA repair and avoid detrimental accumulation of DNA damage. The DNA damage checkpoint can act at multiple stages of the cell cycle to impede DNA replication and block mitotic entry. Initiation of the DNA damage checkpoint signaling relies on two phosphoinositide 3 kinase-related protein kinases, ataxia-telangiectasia mutated (ATM), ATM and RAD3-related (ATR). Upon DNA damage, ATM and

ATR phosphorylate and activate checkpoint kinases CHK1 and CHK2, which, in turn, inhibit CDC25 to prevent activation of cyclin-dependent kinases (CDKs) (*Shiloh, 2003*; *Zhou and Elledge, 2000*).

How does the cell initiate cell cycle recovery after DNA damage is an important, yet largely unanswered, question. Distinct from the simplistic view that the cell cycle resumes passively following the repair of DNA damage and decay of checkpoint signaling, recovery from the DNA damage checkpoint is likely an active and regulated process (*Bartek and Lukas, 2007*; *Clémenson and Marsolier-Kergoat, 2009*). For example, Polo-like kinase 1 (PLK1) and other cell cycle kinases can facilitate the deactivation of DNA damage signaling and resumption of cell cycle progression (*Alvarez-Fernández et al., 2010*; *Macůrek et al., 2008*; *Mamely et al., 2006*; *Peng, 2013*; *Peschiaroli et al., 2006*; *van Vugt et al., 2004a*). Furthermore, it has been observed in yeast, frog, and mammalian cells that the cell cycle can be restarted without completion of DNA repair, a phenomenon defined as adaptation (*Bartek and Lukas, 2007*; *Clémenson and Marsolier-Kergoat, 2009*; *Syljuåsen, 2007*; *Toczyski et al., 1997*; *van Vugt and Medema, 2004b*; *Yoo et al., 2004*). Aside from senescence and other types of permanent cell cycle withdrawal, the DNA damage checkpoint-mediated cell cycle arrest is transient, raising the question about mechanisms that mediate the timely initiation of DNA damage checkpoint recovery.

Microtubule-associated serine/threonine kinase-like (MASTL, also known as Greatwall) has been characterized as an important regulator of mitosis. Like CDK1 and other mitotic kinases, MASTL is activated during mitotic entry, and the kinase activity of MASTL promotes mitosis, although mitotic entry is still permitted in mammalian cells depleted of MASTL (*Archambault et al., 2007*; *Blake-Hodek et al., 2012*; *Castilho et al., 2009*; *Peng and Maller, 2010*; *Vigneron et al., 2011*; *Voets and Wolthuis, 2010*; *Wang et al., 2016*; *Yu et al., 2004*; *Yu et al., 2006*). Upon activation, MASTL phosphorylates α-endosulfine (ENSA) and cyclic AMP-regulated 19 kDa phosphoprotein (ARPP19) which then inhibit PP2A/B55 (protein phosphatase 2A/B55 targeting subunit). Because PP2A/B55 functions as the principal phosphatase catalyzing the dephosphorylation of CDK substrates, the coordination between the kinase activity of MASTL with that of CDK1 enables substrate phosphorylation and mitotic progression (*Castilho et al., 2009*; *Gharbi-Ayachi et al., 2010*; *Mochida et al., 2009*; *Mochida et al., 2010*; *Vigneron et al., 2009*).

Interestingly, in addition to its mitotic function, MASTL is also required for cell cycle recovery from the DNA damage checkpoint in *Xenopus* egg extracts (*Medema, 2010*; *Peng et al., 2011*; *Peng et al., 2010*). Addition of MASTL in extracts facilitated, and depletion of MASTL hindered, DNA damage checkpoint recovery, as evidenced by both mitotic phosphorylation of CDK substrates and de-activation of checkpoint signaling (*Peng et al., 2011*; *Peng et al., 2010*). Consistently, ectopic expression of MASTL in human cells promoted cell proliferation under DNA damage (*Wang et al., 2014*; *Wong et al., 2016*). In this study, we discovered the protein stabilization of MASTL post DNA damage, identified E6 associated protein (E6AP) as the underlying E3 ubiquitin ligase mediating MASTL proteolysis, and delineated ATM-mediated E6AP phosphorylation as a mechanism to initiate DNA damage checkpoint recovery.

## Results

### MASTL expression is upregulated after DNA damage

While investigating the role of MASTL in cell cycle regulation and DDR, we observed unexpected upregulation of MASTL protein levels in cells treated with DNA damage. As shown in *Figure 1A–C*, MASTL upregulation was evident in HEK293 cells treated with doxorubicin (DOX), hydroxyurea (HU), or camptothecin (CPT), generally within hours post treatment. MASTL accumulation was also confirmed in SCC38 cells treated with HU or ionizing radiation (IR, *Figure 1—figure supplement 1A and B*), and HeLa cells treated with DOX or HU (*Figure 1—figure supplement 1C-E*). DNA damage-induced MASTL upregulation was consistent with the activation of ATM/ATR signaling (*Figure 1D and E*), and was not likely to be caused by the completion of DNA repair, as judged by the phosphorylation of replication protein A (RPA, *Figure 1A–D*), H2AX and ATM/ATR substrates (*Figure 1D and E*). In contrast to MASTL, DNA damage did not induce upregulation of other cell cycle kinases that mediate mitotic progression, including CDK1/cyclin B, Aurora A, Aurora B, and PLK1, in HeLa cells treated with DOX (*Figure 1F*), HU (*Figure 1—figure supplement 2A*), or SCC38 cells with HU (*Figure 1—figure supplement 2B*). Furthermore, HU-induced MASTL upregulation was abrogated when cells were

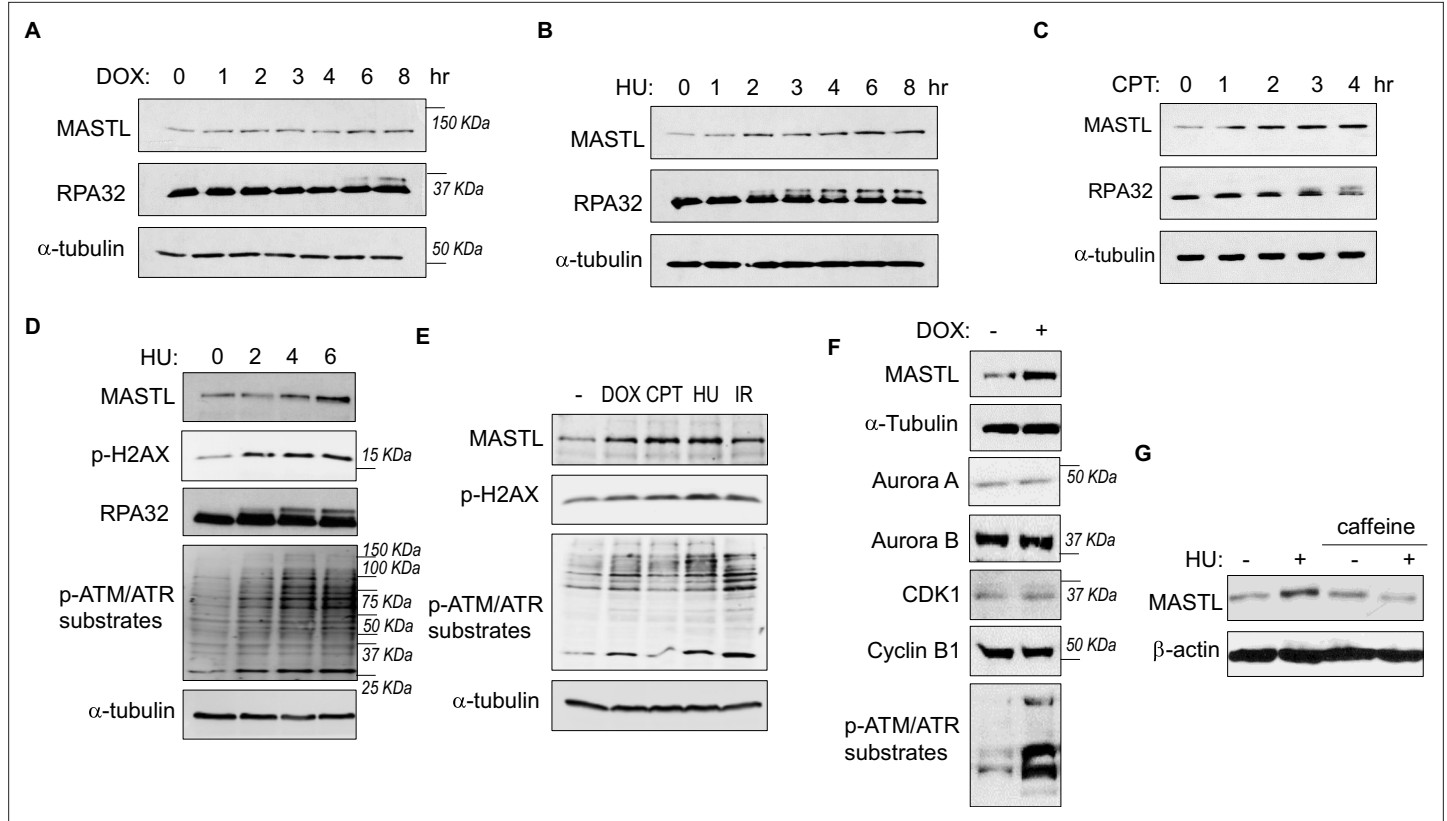

**Figure 1.** The protein level of MASTL is upregulated after DNA damage. (**A**) HEK293 cells were treated with 0.5 μM doxorubicin (DOX) as indicated, cell lysates were collected and analyzed by immunoblotting for MASTL, RPA32, and α-tubulin. (**B**) HEK293 cells were treated with 10 mM hydroxyurea (HU) as indicated, cell lysates were collected and analyzed by immunoblotting for MASTL, RPA32, and α-tubulin. (**C**) HEK293 cells were treated with 10 nM camptothecin (CPT) as indicated, cell lysates were collected and analyzed by immunoblotting for MASTL, RPA32, and α-tubulin. (**D**) HeLa cells were treated with 10 mM HU, and incubated as indicated. Cell lysates were collected and analyzed by immunoblotting for MASTL, phospho-H2AX Ser-139, RPA32, phospho-ATM/ATR substrates, and α-tubulin. (**E**) HeLa cells were treated with or without 0.5 μM DOX, 10 nM CPT, 10 mM HU, and 20 Gy ionizing radiation (IR) for 4 hr. Cell lysates were collected and analyzed by immunoblotting for MASTL, phospho-H2AX Ser-139, phospho-ATM/ATR substrates, and α-tubulin. (**F**) HeLa cells were treated with 0.5 μM DOX, and analyzed by immunoblotting for MASTL, α-tubulin, Aurora A, Aurora B, CDK1, cyclin B1, and phospho-ATM/ATR substrates. (**G**) SCC38 cells were incubated with or without HU and caffeine, as indicated, and analyzed by immunoblotting for MASTL and β-actin.

The online version of this article includes the following figure supplement(s) for figure 1:

**Figure supplement 1.** MASTL upregulation after DNA damage.

**Figure supplement 2.** The expression levels of cell cycle kinases after DNA damage.

treated with caffeine (*Figure 1G*), indicating that DNA damage-induced ATM/ATR activation acted upstream of MASTL upregulation.

## MASTL upregulation after DNA damage is caused by protein stabilization

We did not observe a substantial increase in MASTL RNA transcripts, suggesting MASTL regulation at the post-translational level (*Figure 2A*). Along this line, exogenous MASTL expressed from a different promoter underwent a similar pattern of upregulation after HU, as shown by immunoblotting (*Figure 2B*), and by direct fluorescence (*Figure 2—figure supplement 1A and B*). Like the endogenous MASTL, accumulation of exogenous MASTL occurred concurrently with activation of DNA damage signaling, measured by phosphorylation of ATM/ATR substrates (*Figure 2—figure supplement 1A-C*). These observations prompted us to examine the possibility that DNA damage increased the protein stability of MASTL. Indeed, CPT treatment in HeLa cells increased the protein stability of MASTL, as evaluated by the rate of protein degradation in the presence of cycloheximide (CHX), a

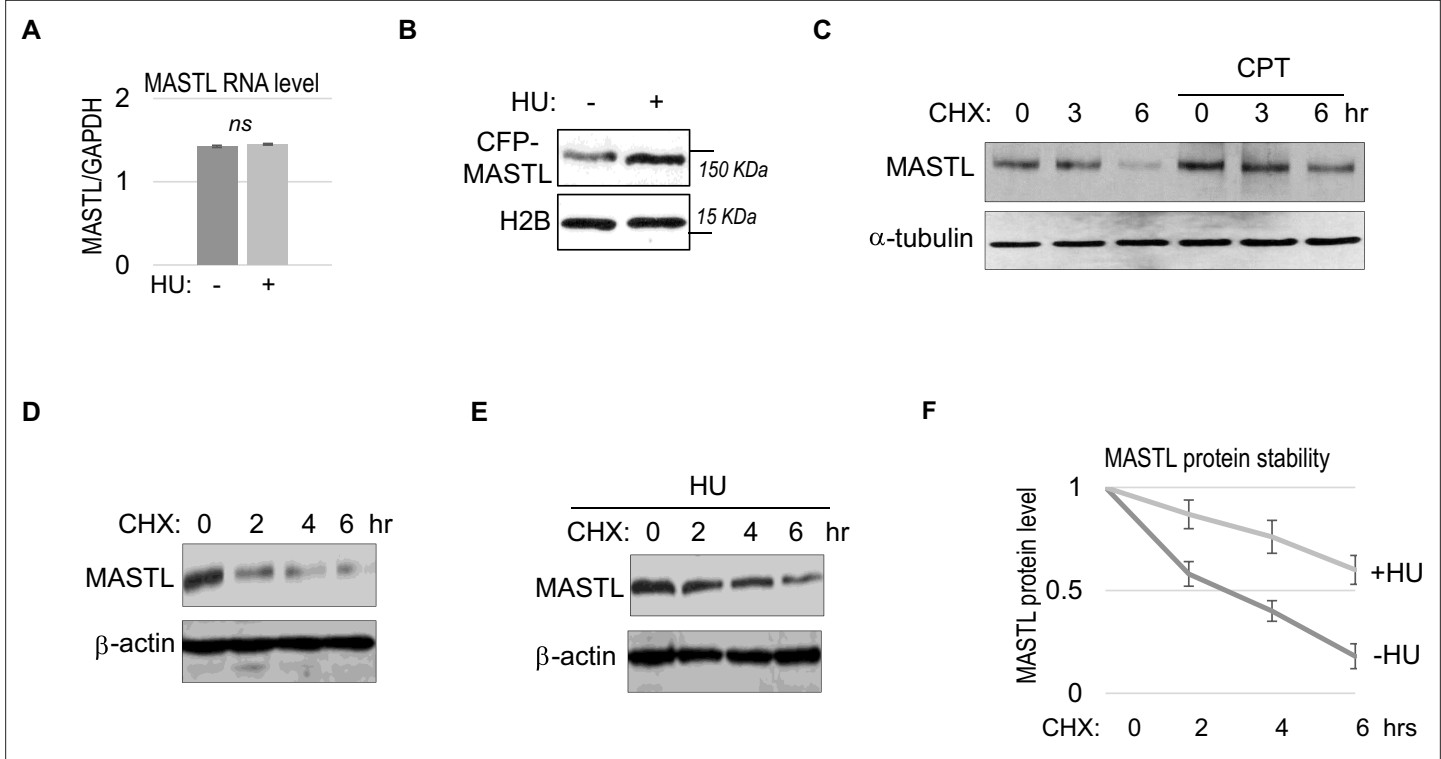

**Figure 2.** MASTL upregulation after DNA damage is mediated by protein stabilization. (**A**) HeLa cells were treated with 10 mM hydroxyurea (HU) for 2 hr, and harvested for gene expression analysis. Quantitative RT-PCR was performed to detect the RNA levels of MASTL and GAPDH. The ratio of MASTL to GAPDH expression is shown. The mean values were calculated from three experiments, and statistical significance was evaluated using an unpaired two-tailed Student's t test. A p-value more than 0.05 was considered non-significant (ns). (**B**) CFP-tagged MASTL was expressed in SCC38 cells. Cells were treated with or without 10 mM HU for 3 hr and analyzed by immunoblotting for CFP-MASTL and H2B. (**C**) HeLa cells were treated with or without 10 nM CPT for 1 hr. These cells were then treated with cycloheximide (CHX, 20 µg/ml) at time 0 to block protein synthesis, and analyzed by immunoblotting for the protein stability of MASTL and α-tubulin. (**D–F**) SCC38 cells were treated without (**D**) or with (**E**) 10 mM HU for 2 hr. These cells were then treated with CHX (20 µg/ml) at time 0 to block protein synthesis, and analyzed by immunoblotting for the protein stability of MASTL and β-actin. In panel F, the band signals were quantified using ImageJ, and the mean values and standard deviations of MASTL/β-actin were calculated based on results of three experiments.

The online version of this article includes the following figure supplement(s) for figure 2:

**Figure supplement 1.** Increased protein stability of MASTL after DNA damage.

protein synthesis inhibitor (*Figure 2C*). Consistently, HU treatment prolonged the half-life of MASTL protein in SCC38 and HEK293 cells (*Figure 2D–F*, *Figure 2—figure supplement 1D and E*).

## E6AP associates with MASTL

To elucidate MASTL regulation via protein stability, we sought to identify the ubiquitin ligase that mediates MASTL proteolysis. We previously performed a proteomic analysis of proteins associated with MASTL (*Ren et al., 2017*), and revealed E6AP as a potential binding partner of MASTL. E6AP is encoded by gene ubiquitin-protein ligase E3A (UBE3A), and is the founding member of the HECT (homologous to E6AP C-terminus)-domain ubiquitin ligase family (*Bernassola et al., 2008*; *Scheffner and Kumar, 2014*). The association between E6AP and MASTL was confirmed by co-immunoprecipitation (*Figure 3A*), and by pulldown of MASTL in HeLa cell lysate using recombinant E6AP (*Figure 3B*). A direct interaction between purified E6AP and MASTL proteins was also evident (*Figure 3—figure supplement 1A*). Further analyses using various segments of MASTL showed that the N-terminal region of MASTL mediated E6AP interaction in HeLa cells and *Xenopus* egg extracts (*Figure 3—figure supplement 1B*, *Figure 3C*). On the other hand, the N-terminus of E6AP co-immunoprecipitated MASTL (*Figure 3D*); the MASTL-binding region was further mapped to aa 208–280 within the N-terminus of E6AP (*Figure 3—figure supplement 1C*). Taken together, our data characterized the

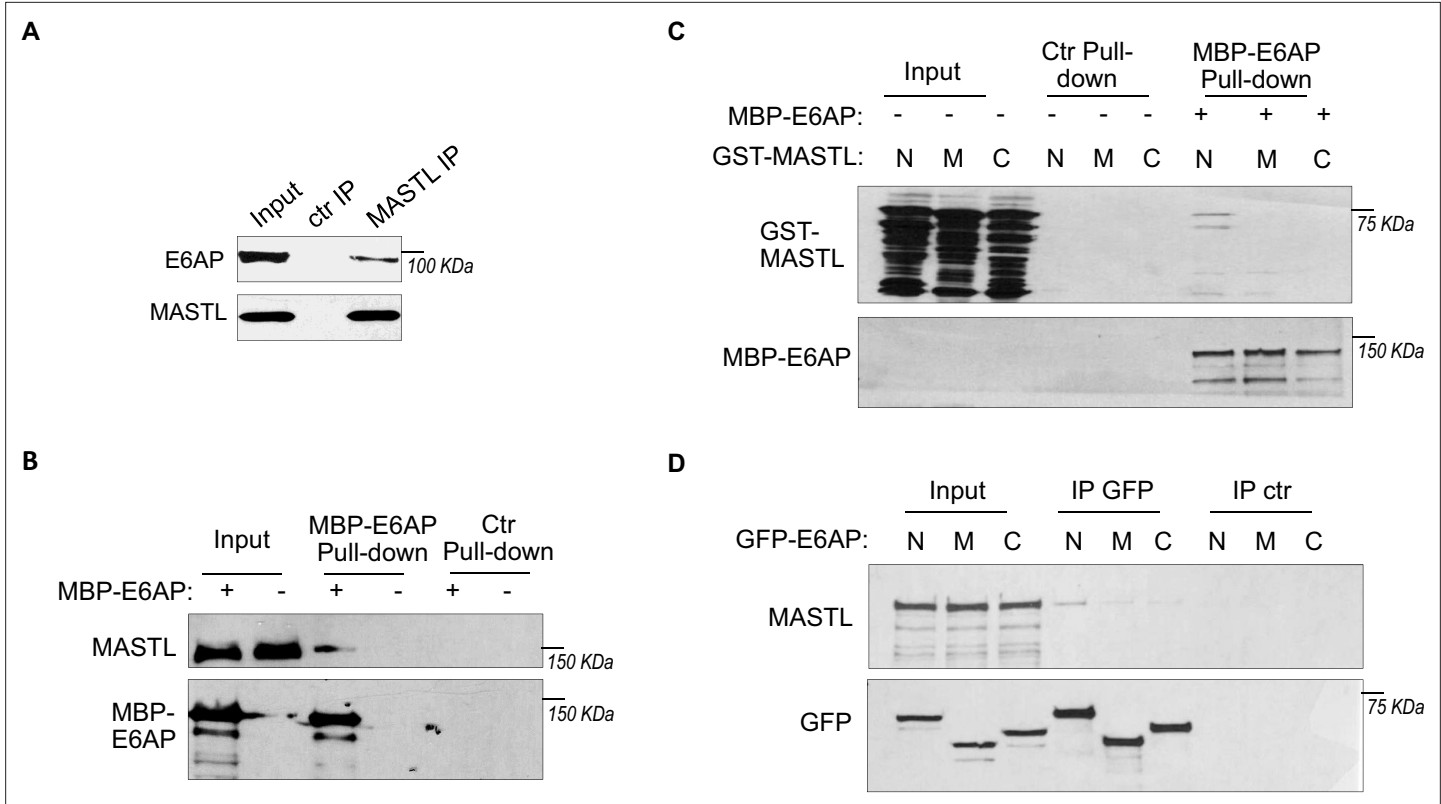

**Figure 3.** E6AP associates with MASTL. (**A**) Immunoprecipitation (IP) was performed in HeLa cell lysates, as described in Materials and methods. The lysate input, MASTL IP, and control (ctr) IP samples were analyzed by immunoblotting for E6AP and MASTL. (**B**) A pulldown assay was performed in HeLa cell lysates using MBP-tagged E6AP, as described in Materials and methods. The pulldown product, cell lysis input, and a control (-) pulldown (using empty beads) were analyzed by immunoblotting for E6AP and MASTL. (**C**) A pulldown assay was performed using MBP-tagged full-length E6AP in *Xenopus* egg extract. Purified segments of MASTL, including N (aa 1–340), M (aa 335–660), and C (aa 656–887), were supplemented in the extracts. The pulldown products, egg extract inputs, and a control (-) pulldown (using empty beads) were analyzed by immunoblotting for GST and MBP. (**D**) Segments of E6AP, including N (aa 1–280), M (aa 280–497), and C (aa 497–770), were tagged with GFP, and transfected into HeLa cells for expression. 24 hr after transfection, cell lysates were harvested for GFP IP. The input, GFP IP, and control (ctr) IP using blank beads were analyzed by immunoblotting for MASTL and GFP.

The online version of this article includes the following figure supplement(s) for figure 3:

**Figure supplement 1.** E6AP and MASTL associate via their N-terminal motifs.

MASTL-E6AP association which was likely mediated via direct protein interaction, although the potential involvement of additional binding partners was not excluded.

## E6AP mediates MASTL ubiquitination and degradation

To determine the potential role of E6AP in MASTL regulation, we depleted E6AP in cells using an siRNA and assessed the mRNA and protein levels of MASTL. As shown in *Figure 4A* and *Figure 4—figure supplement 1A*, E6AP depletion led to an increased level of MASTL protein, without significant change in the MASTL mRNA level. Ectopic expression of E6AP reduced MASTL expression, as shown by immunoblotting (*Figure 4B*) and immunofluorescence (IF) (*Figure 4C*). Consistently, CRISPR-Cas9-mediated gene deletion of E6AP also augmented MASTL level, in a manner that was reversed by re-expression of exogenous E6AP (*Figure 4D*). E6AP depletion increased, and overexpression reduced, the protein stability of MASTL (*Figure 4E and F*). Furthermore, E6AP depletion disrupted MASTL ubiquitination in HeLa cells (*Figure 4G*), indicating that E6AP mediated the ubiquitination of MASTL. Consistently, E6AP depletion did not further increase MASTL level in cells treated with MG132 to suppress proteasome-mediated protein degradation (*Figure 4—figure supplement 1B*). An in vitro ubiquitination assay also confirmed MASTL as an effective substrate of E6AP (*Figure 4H*). MASTL ubiquitination was specifically mediated by E6AP in the assay (*Figure 4—figure supplement*

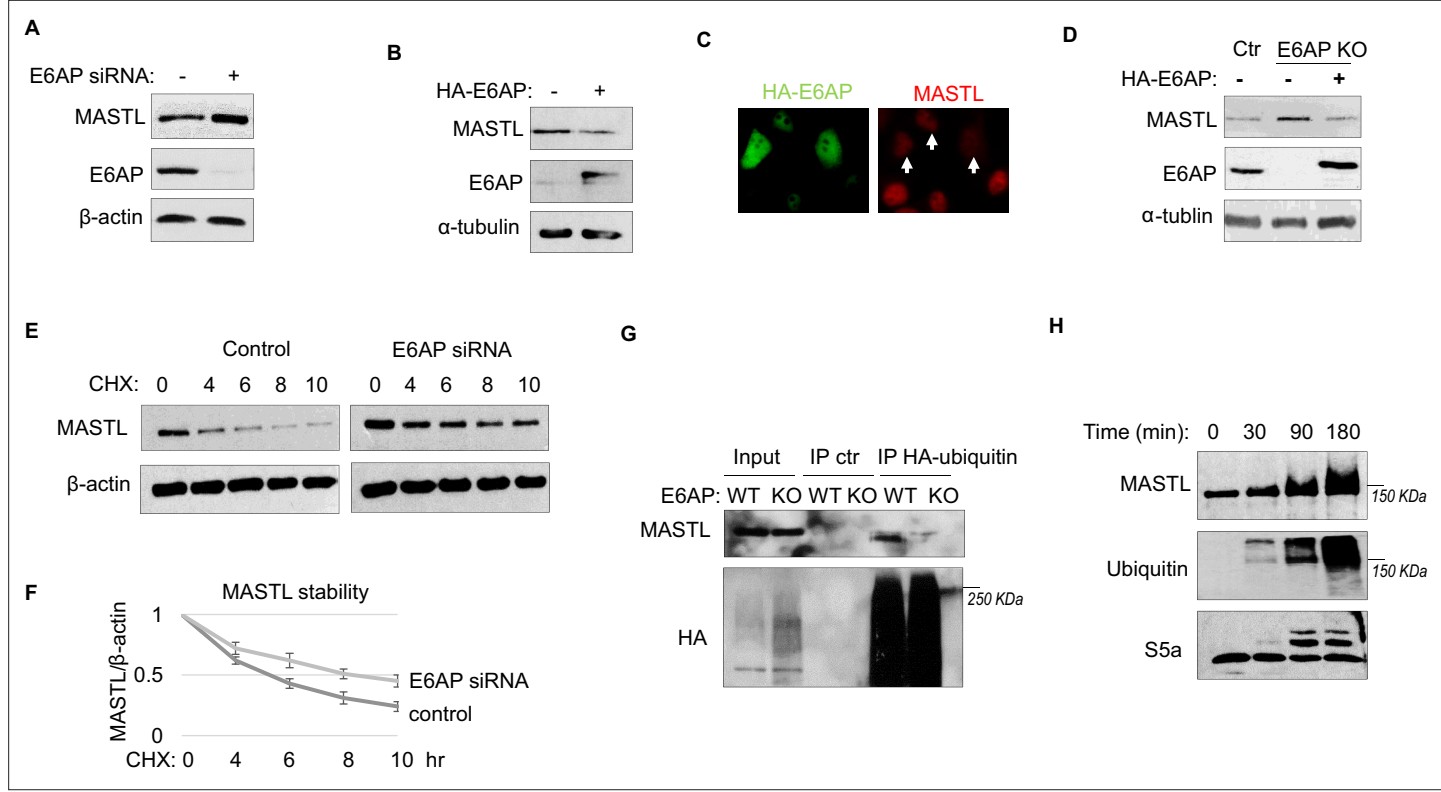

**Figure 4.** E6AP mediates MASTL degradation. (**A**) HeLa cells were transfected with control or E6AP-targeting siRNA E6AP for 24 hr. Cells were analyzed by immunoblotting for E6AP, MASTL, and β-actin. (**B**) HeLa cells were transfected with HA-tagged E6AP. 24 hr after transfection, cells were analyzed by immunoblotting for E6AP, MASLT, and α-tubulin. (**C**) As in panel B, HeLa cells were transfected with HA-E6AP, and analyzed by immunofluorescence (IF) for HA (green) and MASTL (red). Cells with HA-E6AP expression, as denoted by arrowheads, exhibited lower MASTL expression. (**D**) E6AP gene knockout (KO) was performed in HeLa cells, as described in Materials and methods. HA-E6AP was expressed in E6AP KO cells, as indicated. Cells were analyzed by immunoblotting for MASTL, E6AP, and α-tubulin. (**E**) HeLa cells transfected with control or E6AP siRNA were treated with 20 µg/ml cycloheximide (CHX), as indicated. Cells were harvested and analyzed by immunoblotting for MASTL and β-actin. (**F**) HeLa cells were treated as in panel E, MASTL and β-actin protein levels were quantified, and the ratio is shown for the indicated time points after CHX treatment, after normalized to that of time 0. The mean values and standard deviations were calculated from three experiments. (**G**) WT or E6AP KO HeLa cells were transfected with HA-tagged ubiquitin for 12 hr, followed 50 µM MG132 treatment for 4 hr. Cell lysates were harvested for HA immunoprecipitation (IP) or ctr IP using blank beads. The input and IP products were analyzed by immunoblotting for MASTL and HA. (**H**) In vitro ubiquitination assay was performed using E6AP as E3 ligase, and MASTL as substrate, as described in Materials and methods. S5a was added as a control substrate. The reactions were incubated as indicated, as analyzed by immunoblotting for MASTL, ubiquitination, and S5a.

The online version of this article includes the following figure supplement(s) for figure 4:

**Figure supplement 1.** E6AP mediates MASTL ubiquitination and degradation.

1C); the MASTL mutant deleted of the E6AP-binding motif was not ubiquitinated (*Figure 4—figure supplement 1D*). Together, these data established E6AP as a key modulator of MASTL stability.

## E6AP depletion facilitates DNA damage recovery via MASTL

A potential consequence of MASTL accumulation after DNA damage is cell cycle recovery, given the established role of MASTL in promoting cell cycle progression. To analyze cell cycle progression following etoposide (ETO) treatment and release, we quantified mitotic cells that exhibited chromosome condensation and activation of Aurora A/B/C kinases (*Figure 5A*). MASTL depletion substantially hindered DNA damage recovery, whereas E6AP knockout (KO) accelerated mitotic entry after DNA damage (*Figure 5A*). The recovery of mitotic entry in E6AP KO cells was disrupted by MASTL downregulation, indicating its dependence on MASTL (*Figure 5A*). To further study DNA damage recovery, we profiled the cell cycle progression of cells released from HU treatment (*Figure 5B*). MASTL depletion caused prolonged cell cycle arrest; loss of E6AP promoted cell cycle recovery; and suppression of MASTL in E6AP-deleted cells abrogated cell cycle progression (*Figure 5B*). By

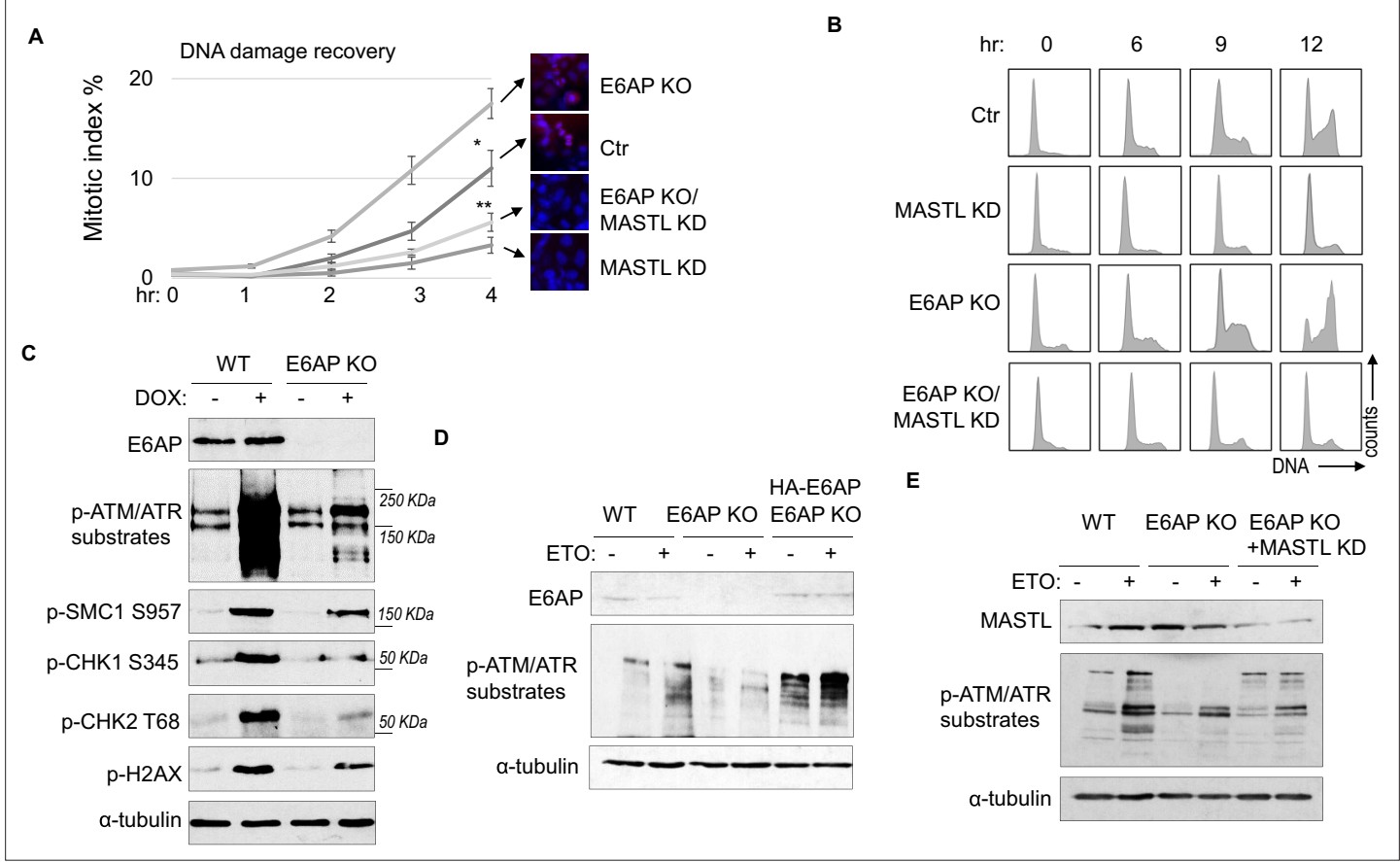

**Figure 5.** E6AP depletion promotes DNA damage checkpoint recovery via MASTL. (**A**) WT or E6AP knockout (KO) HeLa cells were treated with or without MASTL siRNA. The cells were incubated in 0.1 μM etoposide (ETO) for 18 hr, and released in fresh medium for recovery. Cells were harvested at the indicated time points (after the removal of ETO) for immunofluorescence (IF) using an anti-phospho-Aurora A/B/C antibody. The activation of Aurora phosphorylation (shown in red) and chromosome condensation (in blue) indicated mitosis. The percentages of cells in mitosis were quantified manually and shown. The mean values and standard deviations were calculated from three experiments. An unpaired two-tailed Student's t test was used to determine the statistical significance (*p<0.05, **p<0.01, n>500 cell number/measurement). MASTL knockdown by siRNA was shown by immunoblotting in the panel E. (**B**) WT or E6AP KO HeLa cells with or without MASTL siRNA, as in panel A, were treated with 2 mM hydroxyurea (HU) for 18 hr. Cells were then released in fresh medium, and incubated as indicated, for recovery. The cell cycle progression was analyzed by fluorescence-activated cell sorting (FACS), as described in Materials and methods. (**C**) WT or E6AP KO HeLa cells were treated with or without 0.5 μM doxorubicin (DOX) for 4 hr. Cells were then analyzed by immunoblotting for E6AP, phospho-ATM/ATR substrates, phospho-SMC1 Ser-957, phospho-CHK1 Ser-345, phospho-CHK2 Thr-68, γ-H2AX, and α-tubulin. (**D**) WT, E6AP KO, or E6AP KO with expression of HA-E6AP HeLa cells were treated with or without 0.1 μM ETO, and analyzed by immunoblotting for E6AP, phospho-ATM/ATR substrates, and α-tubulin. (**E**) WT, E6AP KO, or E6AP KO with transfection of MASTL siRNA HeLa cells were treated with or without 0.1 μM ETO, and analyzed by immunoblotting for MASTL, phospho-ATM/ATR substrates, and α-tubulin.

The online version of this article includes the following figure supplement(s) for figure 5:

**Figure supplement 1.** Impaired DNA damage checkpoint signaling in E6AP-null cells.

comparison, MASTL knockdown did not significantly disrupt cell proliferation under the unperturbed condition (*Figure 5—figure supplement 1A*).

Interestingly, KO of E6AP in HEK293 cells reduced DNA damage checkpoint signaling, measured by phosphorylation of SMC1, CHK1, CHK2, H2AX, and pan-ATM/ATR substrates after DOX treatment (*Figure 5C*), and by phosphorylation of pan-ATM/ATR substrates after ETO (*Figure 5—figure supplement 1B*). A similar deficiency of ATM/ATR-mediated phosphorylation was observed in HeLa cells with E6AP gene deletion, which was rescued by E6AP re-expression (*Figure 5D*). Interestingly, ATM/ATR-mediated substrate phosphorylation in E6AP KO cells was also restored by MASTL depletion, indicating that E6AP promoted DNA damage checkpoint signaling by counteracting MASTL (*Figure 5E*).

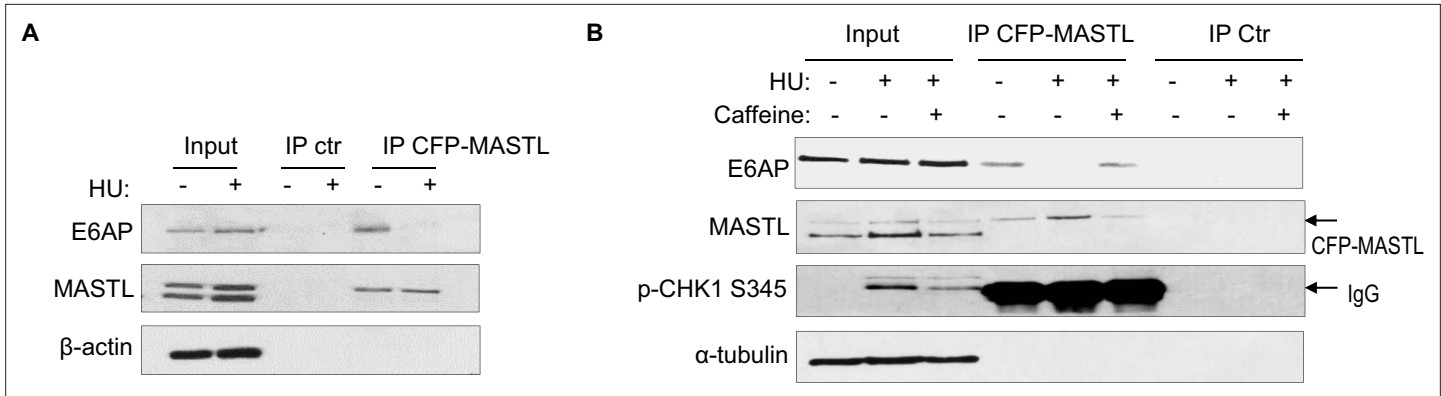

**Figure 6.** The E6AP and MASTL association is disrupted by DNA damage-induced ATM/ATR signaling. (**A**) HeLa cells expressing CFP-MASTL were treated without or with 10 mM hydroxyurea (HU) for 2 hr. CFP-MASTL immunoprecipitation (IP) was performed using a GFP antibody. The input, GFP IP, and control (ctr) IP using blank beads were analyzed by immunoblotting for E6AP, MASTL, and β-actin. (**B**) HeLa cells expressing CFP-MASTL were treated without or with 10 mM HU and 4 mM caffeine, as indicated, for 2 hr. CFP-MASTL IP was performed using a GFP antibody. The input, GFP IP, and control (ctr) IP using blank beads were analyzed by immunoblotting for E6AP, MASTL, phospho-CHK1 Ser-345, and α-tubulin.

## E6AP and MASTL association is regulated by ATM/ATR-mediated DNA damage signaling

Prompted by the findings that E6AP mediated the proteolysis of MASTL and that MASTL protein accumulated following DNA damage, we asked if the association between E6AP and MASTL was impacted by DNA damage. Interestingly, co-immunoprecipitation of E6AP with CPF-MASTL was profoundly disrupted by HU treatment (*Figure 6A*). This loss of E6AP and MASTL association was a result of active DNA damage signaling, as inhibition of ATM/ATR by caffeine preserved this protein association in the presence of HU (*Figure 6B*). These lines of evidence pointed to a model that DNA damage-induced ATM/ATR activation disrupts E6AP association with MASTL, subsequently leading to reduced MASTL degradation and increased MASTL protein accumulation.

## ATM/ATR mediates E6AP S218 phosphorylation to modulate E6AP and MASTL association

ATM/ATR phosphorylates numerous substrates to regulate their functions after DNA damage. These kinases typically target serine or threonine residues followed by glutamine (S/TQ). E6AP possesses a single evolutionarily conserved serine, Ser-218 in humans, as a potential consensus site of ATM/ATR-mediated phosphorylation (*Figure 7A*). E6AP Ser-218 phosphorylation was also documented in multiple proteomic databases (*Beli et al., 2012*; *Franz-Wachtel et al., 2012*; *Kettenbach et al., 2011*; *Klammer et al., 2012*; *Matsuoka et al., 2007*; *Olsen et al., 2010*; *Schweppe et al., 2013*; *Weber et al., 2012*). We generated a phospho-specific antibody for this residue, and observed the induction of Ser-218 phosphorylation after HU (*Figure 7B*), or DOX treatment (*Figure 7C*). This phospho-signal was absent in E6AP KO cells (*Figure 7B*, *Figure 7—figure supplement 1A*), and was diminished with S218A mutation (*Figure 7E*), confirming its specificity. Inhibition of ATM using a selective kinase inhibitor reduced E6AP Ser-218 phosphorylation after DOX treatment in both HeLa and HEK293 cells (*Figure 7C*, *Figure 7—figure supplement 1B*). ATM was also the primary kinase to mediate E6AP Ser-218 phosphorylation in response to HU (*Figure 7—figure supplement 1C*).

As we showed E6AP-MASTL dissociation in the DNA damage-induced and ATM/ATR-dependent manner, we hypothesized that E6AP phosphorylation modulated its protein association with MASTL. Indeed, the phospho-mimetic mutant form of E6AP, E6AP S218D, exhibited significantly reduced association with MASTL, compared to WT or phospho-deficient S218A E6AP (*Figure 7D*). Furthermore, DNA damage disrupted MASTL association with WT E6AP, but not E6AP S218A (*Figure 7E*). Consistent with the persistent MASTL association, S218A mutation also prevented MASTL protein accumulation after DNA damage (*Figure 7F*).

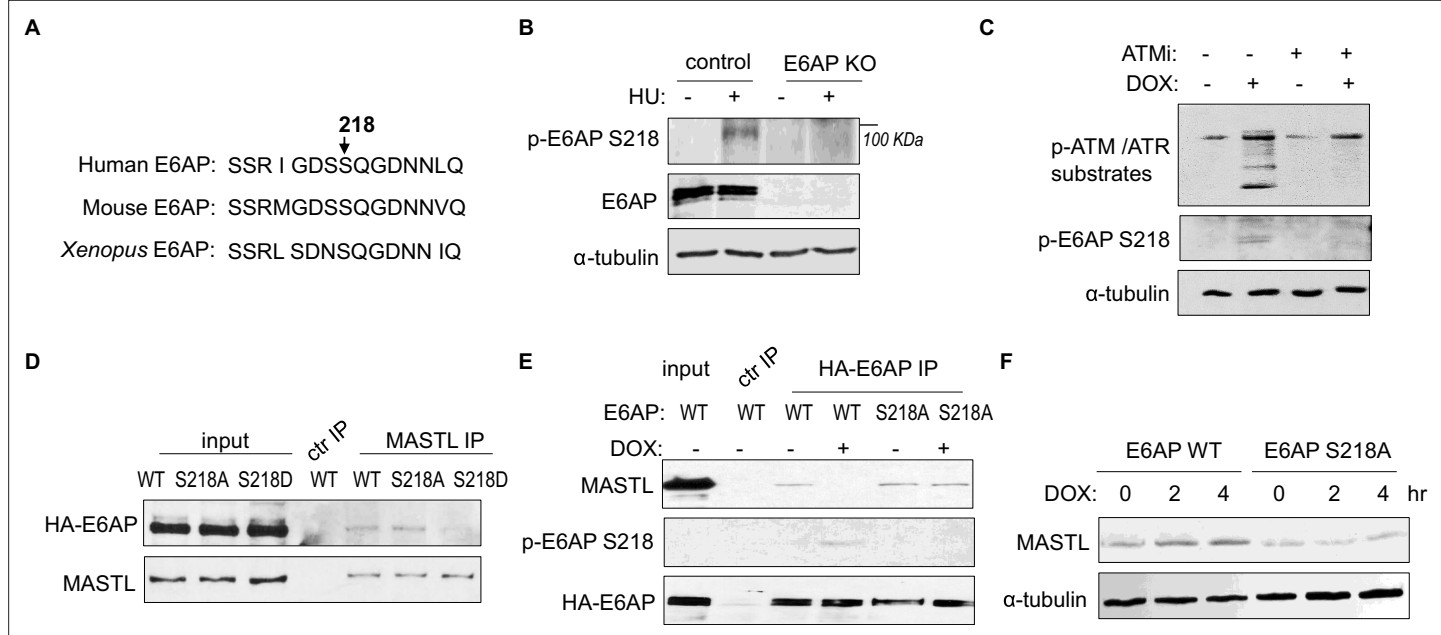

**Figure 7.** ATM/ATR mediates E6AP S218 phosphorylation to disrupt MASTL association and proteolysis. (**A**) The sequence alignment of the conserved E6AP Ser-218 motif in human, mouse, and *Xenopus*. (**B**) A phospho-specific antibody recognizing E6AP Ser-218 was generated, as described in Materials and methods. WT or E6AP knockout (KO) HEK293 cells were treated without or with 10 mM hydroxyurea (HU) for 4 hr, and analyzed by immunoblotting for phospho-E6AP Ser-218, E6AP, and α-tubulin. (**C**) HeLa cells were treated without or with 0.5 μM doxorubicin (DOX) and 5 μM KU55933 (ATMi), as indicated, for 1 hr, and analyzed by immunoblotting for phospho-ATM/ATR substrates, phospho-E6AP Ser-218, and α-tubulin. (**D**) HeLa cells were transfected with HA-tagged WT, S218A, or S218D E6AP. Cell lysates were harvested for immunoprecipitation (IP) assays. The input, MASTL IP, and a control IP using empty beads products were analyzed by immunoblotting for MASTL and HA. (**E**) HeLa cells were transfected with HA-tagged WT or S218A E6AP, as in panel D. Cells were treated with or without 0.5 μM DOX for 3 hr, and harvested for IP assays. The input, HA IP, and a control IP using empty beads products were analyzed by immunoblotting for MASTL, phospho-E6AP Ser-218, and HA. (**F**) E6AP KO HeLa cells were transfected with HA-tagged WT or S218A E6AP, as in panel D. Cells were treated with or without 0.5 μM DOX, incubated as indicated, and harvested for immunoblotting for MASTL and α-tubulin.

The online version of this article includes the following figure supplement(s) for figure 7:

**Figure supplement 1.** DNA damage-induced E6AP Ser-218 phosphorylation is mediated by ATM.

## E6AP S218 phosphorylation promotes DNA damage checkpoint recovery

As we established E6AP S218 phosphorylation in response to DNA damage, we sought to investigate the functional consequence of E6AP Ser-218 phosphorylation after DNA damage. Cell cycle recovery was examined in E6AP KO cells reconstituted with either WT or S218A E6AP. Interestingly, using both mitotic index measurement after ETO release, or cell cycle profiling after HU, we noted defective DNA damage checkpoint recovery in cells harboring E6AP S218A mutation (*Figure 8A and B*). The resumption of cell cycle progression post DNA damage was also assessed biochemically, by phosphorylation of CDK substrates, and a similar pattern of deficiency was seen with S218A mutation (*Figure 8C*). Furthermore, cells expressing S218A E6AP, compared to those expressing WT E6AP, exhibited elevated levels of DNA damage signaling (*Figure 8D*).

## Discussion

In this study, we identified E6AP as the underlying E3 ubiquitin ligase that mediated the ubiquitination and degradation of MASTL. Compared to the function of MASTL, regulation of MASTL is relatively underinvestigated. Previous studies illustrated mechanisms that modulated the phosphorylation and subcellular localization of MASTL during cell cycle progression (*Castro and Lorca, 2018*). We reported here that DNA damage disrupted E6AP and MASTL association, leading to increased MASTL protein stability and accumulation of MASTL protein. Because MASTL promotes the phosphorylation of CDK

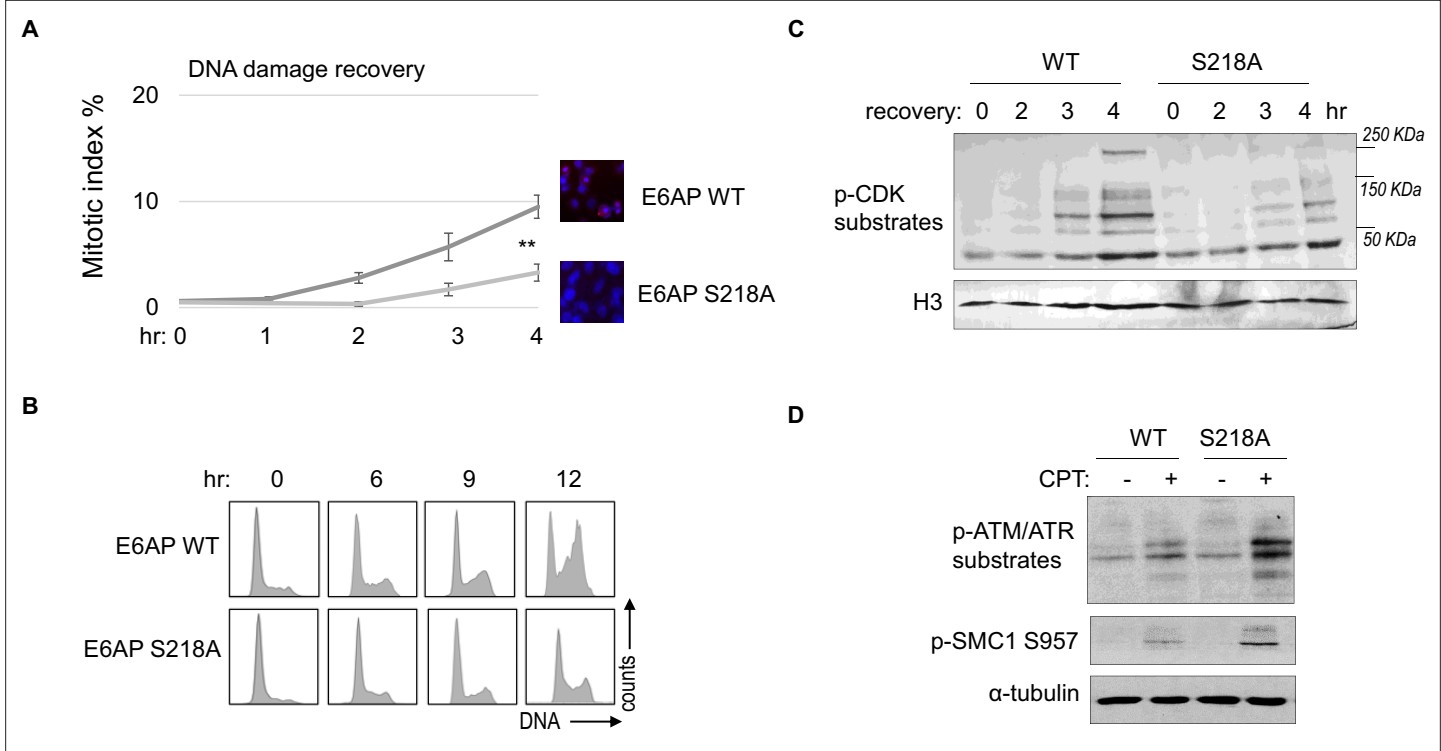

**Figure 8.** E6AP S218 phosphorylation is required for DNA damage recovery. (**A**) E6AP knockout (KO) HeLa cells were transfected with HA-tagged WT or S218A E6AP, as in *Figure 7*. Cells were treated with 0.1 µM etoposide (ETO) for 18 hr, and released in fresh medium for recovery. Cells were then harvested at the indicated time points (after the removal of ETO) for immunofluorescence (IF) using an anti-phospho-Aurora A/B/C antibody. The activation of Aurora phosphorylation (shown in red) and chromosome condensation (in blue) indicated mitosis. The percentages of cells in mitosis were quantified manually and shown. The mean values and standard deviations were calculated from three experiments. An unpaired two-tailed Student's t test was used to determine the statistical significance (**p<0.01, n>500 cell numbers/measurement). (**B**) E6AP KO HeLa cells expressing HA-tagged WT or S218A E6AP, as in panel A, were treated with 2 mM hydroxyurea (HU) for 18 hr. Cells were then released in fresh medium, and incubated as indicated, for recovery. Cell cycle progression was analyzed by fluorescence-activated cell sorting (FACS). (**C**) WT or S218A E6AP was expressed in E6AP KO HEK293 cells. Cells were treated without or with 0.1 µM ETO for 18 hr, released in fresh medium for recovery, and incubated as indicated. Cells were analyzed by immunoblotting for phospho-cyclin-dependent kinase (CDK) substrates and histone H3. (**D**) WT or S218A E6AP was expressed in E6AP KO HEK293 cells, as in panel C. Cells were treated without or with 1 µM CPT for 90 min, and analyzed by immunoblotting for phospho-ATM/ATR substrates, phospho-SMC1 Ser-957, and α-tubulin.

substrates by inhibiting the counteracting phosphatase PP2A/B55, this fashion of MASTL upregulation can re-activate cell cycle progression while CDK activities are restrained by the DNA damage checkpoint (*Figure 9*). By comparison, other cell cycle kinases that promote mitotic progression, including CDK1/cyclin B, Aurora A/B, and PLK1, did not exhibit this pattern of DNA damage-induced upregulation.

E6AP, also known as UBE3A, is the prototype of the E3 ligase subfamily containing a C-terminal HECT domain. Mutations of E6AP cause Angelman syndrome, a debilitating neurological disorder in humans (*Bernassola et al., 2008*; *Buiting et al., 2016*; *Levav-Cohen et al., 2012*; *Sell and Margolis, 2015*; *Wolyniec et al., 2013*). E6AP and other HECT-containing E3 ligases are emerging as potential etiological factors and drug targets in cancer (*Bernassola et al., 2008*; *Scheffner and Kumar, 2014*; *Yu et al., 2020*). Interestingly, mouse embryo fibroblasts lacking E6AP escaped replicative senescence, proliferated under stress conditions, supported anchorage-independent growth, and enhanced tumor growth in vivo (*Levav-Cohen et al., 2012*; *Wolyniec et al., 2013*). These phenotypes are not well explained in the context of known substrates of E6AP (*Sell and Margolis, 2015*), but can be potentially attributed to MASTL upregulation, as characterized in our current study. E6AP binds MASTL via its N-terminal domain, and mediates MASTL ubiquitination and degradation. The function of E6AP in the DNA damage checkpoint via MASTL modulation suggests E6AP as a DDR factor and a potential tumor suppressor.

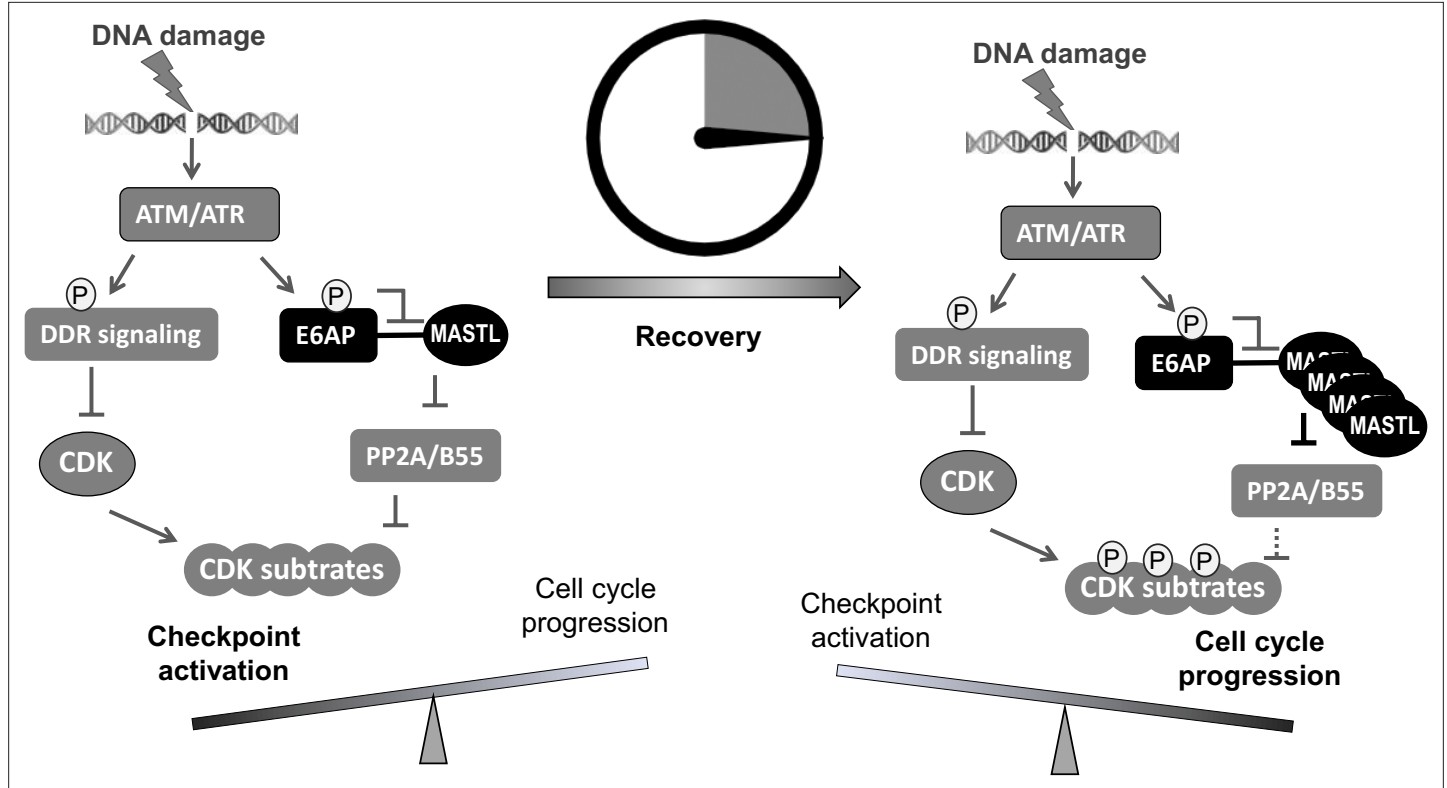

**Figure 9.** A 'timer' model for the role of the ATM-E6AP-MASTL axis in cell cycle arrest and recovery after DNA damage. DNA damage induces ATM/ATR activation and checkpoint signaling. At this stage (the initial DNA damage sensing/signaling), MASTL expression is normal (not yet upregulated). Activated ATM/ATR also phosphorylates E6AP Ser-218, leading to the dissociation of E6AP from MASTL and reduced MASTL degradation. The subsequent accumulation of MASTL, at upregulated levels several hours after DNA damage, promotes de-activation of the DNA damage checkpoint and initiates cell cycle resumption by inhibiting dephosphorylation of cyclin-dependent kinase (CDK) substrates.

ATM/ATR phosphorylates a myriad of substrates to promote DNA repair and activate the cell cycle checkpoints (*Flynn and Zou, 2011*; *Shiloh, 2003*). We found that MASTL accumulation after DNA damage was disrupted by inhibition of ATM/ATR, suggesting that ATM/ATR modulated MASTL proteolysis. Our study further revealed that ATM phosphorylated E6AP at Ser-218, leading to E6AP dissociation from MASTL and the subsequent MASTL stabilization. Our functional characterization of E6AP Ser-218 is interesting, as phosphorylation of this residue has been detected in numerous proteomic studies as a modification induced by DNA damage (*Beli et al., 2012*; *Matsuoka et al., 2007*), mitosis (*Franz-Wachtel et al., 2012*; *Kettenbach et al., 2011*; *Olsen et al., 2010*), or cancer progression (*Klammer et al., 2012*; *Schweppe et al., 2013*; *Weber et al., 2012*).

Our results suggest that ATM/ATR, while important for the activation of the DNA damage checkpoint, also engages a mechanism to initiate cell cycle recovery (*Figure 9*). This mechanism at least partially answers the puzzling question of how cell cycle recovery is initiated from the state of the DNA damage checkpoint. The DNA damage checkpoint targets cell cycle kinases to halt the cell cycle; on the other hand, the re-activation of CDK, PLK1, MASTL, and other cell cycle kinases promotes the de-activation of DNA damage checkpoint signaling and cell cycle resumption (*Peng, 2013*). For example, PLK1 and CDK can mediate the phosphorylation of DDR factors to suppress DNA damage checkpoint signaling (*Alvarez-Fernández et al., 2010*; *Macůrek et al., 2008*; *Mamely et al., 2006*; *Peschiaroli et al., 2006*; *van Vugt et al., 2004a*). Cell cycle kinases are known to coordinate with each other in positive feedback reactions to achieve full activation and bring about mitosis. We speculate that, in the case of DNA damage recovery, ATM-mediated MASTL upregulation may provide an initial signal to trigger these positive feedback reactions, ultimately shifting the balance from cell cycle arrest to recovery. Of note, this timer-like mechanism can contribute to the transient nature of the DNA damage checkpoint, as observed in yeast, frog, and mammalian cells (*Bartek and Lukas, 2007*; *Clémenson and Marsolier-Kergoat, 2009*; *Syljuåsen, 2007*; *van Vugt et al., 2004a*; *Yoo*

*et al., 2004*). In comparison to the recovery stage, the expression level of MASTL is not upregulated during the activation stage of the DNA damage checkpoint, allowing efficient checkpoint activation. The subsequent impact of MASTL upregulation on promoting checkpoint recovery and cell cycle progression can be attributed to inhibition of PP2A/B55, although the potential involvement of additional mechanisms is not excluded. Finally, the resumption of the cell cycle after DNA damage is a potentially vital process that enables tumor cell progression and treatment evasion. MASTL kinase has emerged as an important factor of tumorigenesis and treatment resistance in multiple types of cancer (*Fatima et al., 2020*; *Marzec and Burgess, 2018*; *Wang et al., 2014*). Thus, future studies built upon our current findings may shed new light on cancer resistance and identify new anti-cancer drug targets to enhance treatment outcome.

## Materials and methods

### Antibodies and chemicals

Mouse antibody to MASTL (clone 4F9, Millipore MABT372) was described previously (*Wang et al., 2011*). Phospho-specific E6AP Ser-218 antibody was generated using a synthesized peptide (SSRIGDS phospho-S QGDNNLQ). Other antibodies include α-tubulin (Santa Cruz Biotechnology, #sc-5286), E6AP (Bethyl Laboratories, A300-351), HA (Cell Signaling Technology #3724), γ-H2AX Ser-139 (Cell Signaling Technology #9718S), phospho-ATM/ATR substrate motif (Cell Signaling Technology, #6966S), phospho-SMC1 Ser-957 (Cell Signaling Technology, #58052), phospho-CHK1 Ser-345 (Cell Signaling Technology, #2348), phospho-CHK2 Thr-68 (Cell Signaling Technology, #2197), Phospho-Aurora A (Thr288)/Aurora B (Thr232)/Aurora C (Thr198) (Cell Signaling Technology, #2914), Aurora A (Cell Signaling Technology, #14475), Aurora B (Cell Signaling Technology, #3094), CDK1 (Cell Signaling Technology, #9112), Cyclin B (Cell Signaling Technology, #4138), phosphor-CDK substrates (Cell Signaling Technology, #2325), RPA32 (Thermo Fisher Scientific, #PA5-22256), S5a (Boston Biochem, #SP-400), ubiquitin (Cell Signaling Technology, #3936), and GFP (Cell Signaling Technology, #2555).

The following chemicals were used: HU (MP Biomedicals, #102023), DOX (MilliporeSigma, #25316-40-9), caffeine (Sigma-Aldrich, #C0750), ATM inhibitor (KU55933, Selleckchem, #S1092), ATR inhibitor (VE-821, Selleckchem, #S8007), CHX (Fluka analytical, #01810), ETO (Sigma-Aldrich, #E1383), CPT (Sigma-Aldrich, #C9911), G418 sulfate (Thermo Fisher Scientific, #10131035), MG132 (Calbiochem, #133407-82-6), cisplatin (R&D Systems, #15663-27-1), propidium iodide (PI, Thermo Fisher Scientific, #P1304MP), and isopropyl-beta-D-thiogalactopyranoside (IPTG, RPI research products international, #367-93-1).

### Cell culture and treatment

Human cervix carcinoma (HeLa) and human embryonic kidney 293 (HEK293) cell lines were obtained and authenticated by ATCC, and maintained in Dulbecco's modified Eagle medium (DMEM, Hyclone) with 10% fetal bovine serum (FBS, Hyclone). Human head and neck squamous cell carcinoma UM-SCC-38 cells, as characterized in *Wang et al., 2014*, was maintained in DMEM (HyClone) with 10% FBS (HyClone). As described previously (*Li et al., 2021*), transfection of plasmid vectors was carried out using Lipofectamine 2000 (Invitrogen) following the manufacturer's protocol. siRNA targeting human UBE3A or human MASTL (Integrated DNA Technologies) was transfected into cells using Lipofectamine RNAi MAX (Invitrogen). A non-targeting control siRNA was used as a control. UBE3A siRNA sequence: 5'-3'AGGAAUUUGUCAAUCUUU; 5'-3' UCAGAAUAAAGAUUGACA. MASTL siRNA sequence: 5'-3'GUCUACUUGGUAAUGGAA; 5'-3' UAAGAUAUUCCAUUACCA. For fluorescence-activated cell sorting, cells were fixed in 70% cold ethanol, washed with cold PBS, and stained with propidium iodide (20 μg/ml propidium iodide and 200 μg/ml RNAse A diluted in PBS with 0.1% Triton X-100) at 37°C for 15 min before analysis using BD FACSArray.

### Generation of E6AP KO cells

To generate E6AP KO HeLa cells, a pCRSIPRv2-sgRNA construct expressing both Cas9 and an sgRNA targeting human E6AP were transfected into HeLa cells. Twenty-four hours after transfection, cells were selected with puromycin (1.5 μg/ml) for 2 days. Single cells were grown in 96-well plates for amplification. Individual clones were verified by immunoblotting for E6AP expression. The following sgRNA sequence was used for gene KO: 5' CTACTACCACCAGTTAACTG 3'. E6AP KO was also

carried out in HEK293 cells using a CRISPRevolution sgRNA EZ Kit (Synthego), following the manufacturer's protocol. The following sgRNA sequence was used for gene KO: 5' GCAAGCTGACACAGGT GCTG 3'.

## Immunoblotting, IF, and immunoprecipitation

Immunoblotting, IF, and immunoprecipitation (IP) were performed as previously described (*Wang et al., 2019*). Briefly, for immunoblotting, samples were harvested in 1X Laemmli sample buffer (Bio-Rad) and resolved by sodium dodecyl sulfate-polyacrylamide gel electrophoresis. After electro-transfer, PVDF membranes (Millipore, Billerica, MA, USA) were blocked in 1× TBST (10 mM Tris-HCl, pH 7.5, 150 mM NaCl, 0.05% Tween 20) containing 5% nonfat dry milk. Membranes were incubated in primary antibodies in a primary antibody dilution buffer (1X TBS, 0.1% Tween-20 with 5% BSA), and then horseradish peroxidase-conjugated secondary antibodies (Sigma) in 1× TBST. Detection was performed using an enhanced chemiluminescence substrate kit (Thermo Scientific Pierce).

For IF, cells on microscope cover glasses were washed with PBS, fixed in 3% formaldehyde with 0.1% Triton X-100, permeabilized in 0.05% saponin, and blocked with 5% goat serum. Primary antibodies were diluted in the blocking buffer and incubated with the cells for 2 hr. The cells were then incubated with Alexa Fluor secondary antibodies (Invitrogen, 1: 2000) for 1 hr at room temperature. The nuclei of cells were stained with 4',6-diamidino-2-phenylindole. Imaging was performed using a Zeiss Axiovert 200M inverted fluorescence microscope at the UNMC Advanced Microscopy Core Facility.

For IP, cells were harvested in lysis 150 buffer (50 mM HEPES [pH 7.5], 150 mM NaCl, 1 mM DTT, and 0.5% Tween 20). Anti-rabbit or anti-mouse magnetic beads (Thermo Fisher) were conjugated to antibodies, and incubated in cell lysates for IP.

## In vitro ubiquitination assay

The in vitro ubiquitination assay was performed using an ubiquitin kit (Boston Biochem, #K-230). His-tagged S5a protein (provided in the kit as positive control) or GST-tagged MASTL protein (purified as below) was added in the ubiquitin reactions as substrate. After incubation at 37°C for up to 180 min, the reactions were terminated by the addition of Laemmli buffer and boiling.

## Plasmid construction and protein expression

A vector expressing HA-tagged human E6AP was obtained from Addgene (Plasmid #8658), E6AP mutants were generated using site-directed mutagenesis (Agilent) following the protocol recommended by the manufacturer. Segments of E6AP, including N (aa 1–280), M (aa 280–497), C (aa 497–770), were inserted to a pEGFP vector for IP, and pMBP vector for pulldown. Additional segments, including N1 (aa 1–99), N2 (aa 100–207), and N3 (108-280), were cloned to pEGFP for IP. Expression vectors for MASTL were previously characterized (*Yamamoto et al., 2014*). Additionally, three segments of MASTL (N: aa 1–340; M: aa 335–660; C: aa 656–887) were inserted into pGEX 4T-1 (GE Healthcare). The resulting expression vectors were transfected into BL21 bacteria cells for protein expression and purification. For the pulldown experiments, GST-tagged proteins were purified on Glutathione Sepharose beads (New England Biolabs), and MBP-tagged proteins were purified on Amylose resin (New England Biolabs).

## *Xenopus* egg extracts

Cytostatic factor extracts were prepared as described previously (*Zhu and Peng, 2016*). Eggs were incubated in 2% cysteine and washed in 1× XB (1 M KCl, 11 mM MgCl$_2$, 100 mM HEPES [pH 7.7], and 500 mM sucrose). Egg extracts were generated by centrifugation at 10,000 × g. Extracts were released into interphase by supplementation with 0.4 mm CaCl$_2$, and incubated for 30 min at room temperature.

## Acknowledgements

We thank Drs. Gregory G Oakley, Thomas M Petro, and Dr. Jixin Dong (University of Nebraska Medical Center, USA) for stimulating discussions. AP is supported by funding from the National Institutes of Health (CA233037; DE030427).

## Additional information

### Funding

| Funder | Grant reference number | Author |
| --- | --- | --- |
| National Institutes of Health | CA233037 | Aimin Peng |
| National Institutes of Health | DE030427 | Aimin Peng |

The funders had no role in study design, data collection and interpretation, or the decision to submit the work for publication.

### Author contributions

Yanqiu Li, Conceptualization, Data curation, Formal analysis, Validation, Investigation, Writing - original draft; Feifei Wang, Conceptualization, Data curation, Formal analysis, Investigation; Xin Li, Data curation, Investigation; Ling Wang, Data curation, Formal analysis, Investigation; Zheng Yang, Resources; Zhongsheng You, Resources, Writing - review and editing; Aimin Peng, Conceptualization, Formal analysis, Supervision, Funding acquisition, Investigation, Writing - original draft, Writing - review and editing

### Author ORCIDs

Xin Li ⓘ http://orcid.org/0000-0002-5200-3858
Zhongsheng You ⓘ http://orcid.org/0000-0002-9719-8791
Aimin Peng ⓘ http://orcid.org/0000-0002-2452-1949

Reviewer #1 (Public Review): https://doi.org/10.7554/eLife.86976.3.sa1
Reviewer #2 (Public Review): https://doi.org/10.7554/eLife.86976.3.sa2
Author Response https://doi.org/10.7554/eLife.86976.3.sa3

## Additional files

### Supplementary files

• Source data 1. Original gel blots, as presented in the figures and figure supplements, are provided as source data.

### Data availability

All data generated or analyzed during this study are included in the manuscript and supporting file.

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
