## [Editor Report · eLife assessment]

This study reports the **important** finding that there appears to be a timer that monitors the repair of DNA after damage and regulates whether cells are subsequently able to enter mitosis. The authors identify proteins important for this decision and propose a mechanism supported by **solid** but not conclusive data. This study will be of interest to researchers in the fields of DNA damage repair and cell cycle control.

---

## [Referee Report · Reviewer #1 (Public Review)]

In principle a very interesting story, in which the authors attempt to shed light on an intriguing and poorly understood phenomenon; the link between damage repair and cell cycle re-entry once a cell has suffered from DNA damage. The issue is highly relevant to our understanding of how genome stability is maintained or compromised when our genome is damaged. The authors present the intriguing conclusion that this is based on a timer, implying that the outcome of a damaging insult is somewhat of a lottery; if a cell can fix the damage within the allocated time provided by the "timer" it will maintain stability, if not then stability is compromised. If this conclusion can be supported by solid data, the paper would make a very important contribution to the field.

However, the story in its present form suffers from a number of major gaps that will need to be addressed before we can conclude that MASTL is the "timer" that is proposed here. The primary concern being that altered MASTL regulation seems to be doing much more than simply acting as a timer in control of recovery after DNA damage. There is data presented to suggest that MASTL directly controls checkpoint activation, which is very different from acting as a timer. The authors conclude on page 8 "E6AP promoted DNA damage checkpoint signaling by counteracting MASTL", but in the abstract the conclusion is "E6AP depletion promoted cell cycle recovery from the DNA damage checkpoint, in a MASTL-dependent manner". These 2 conclusions are definitely not in alignment. Do E6AP/MASTL control checkpoint signaling or do they control recovery, which is it?

Also, there is data presented that suggest that MASTL does more than just controlling mitotic entry after DNA damage, while the conclusions of the paper are entirely based on the assumption that MASTL merely acts as a driver of mitotic entry, with E6AP in control of its levels. This issue will need to be resolved.

and finally, the authors have shown some very compelling data on the phosphorylation of E6AP by ATM/ATR, and its role in the DNA damage response. But the time resolution of these effects in relation to arrest and recovery have not been addressed.

Revised manuscript:

I think the authors did a good job in revising the paper, and provide compelling support for a timer function in the checkpoint. I do think they still have missed one important point how MASTL could act as a timer to control recovery. The data clearly show that MASTL somehow controls ATM/ATR activity, whilst their final model (fig.9) places MASTL upstream of CDK activity, without mentioning its feedback on ATM/ATR. I think there are 2 possible explanations for the timer function of MASTL they have discovered here, both may be relevant. The first is enhanced CDK activation by direct control of CDK phosphorylation through MASTL/B55/PP2A. The second is through MASTL-mediated shut-down of ATM/ATR activation (mechanism to be determined) which is also reported here. Their final model and discussion do not display sufficient appreciation for this latter option, and I would argue that the HU-recovery experiment shown in Fig.5B is actually in strong support of the second explanation, rather than the first.

---

## [Referee Report · Reviewer #2 (Public Review)]

This manuscript describes a role for the ATM-E6AP-MASTL pathway in DNA damage checkpoint recovery. However, the data in the first version of the manuscript strongly suggest that E6AP is involved in checkpoint activation, which raises doubts about the exact function of this pathway. Additional minor issues were raised regarding the quality of some of the data. Although some minor points were addressed in the revised manuscript, the major issue whether the E6AP-MASTL pathway mediates checkpoint activation or checkpoint recovery was not experimentally addressed. Instead, the authors state that "the expression level of MASTL is not upregulated during the activation stages of the DNA damage checkpoint". However, their data show otherwise: MASTL upregulation coinciding with RPA phosphorylation and p-ATM/ATR signal.

I am therefore not convinced the revised manuscript sufficiently addressed the comments to fully support the conclusions.

---

## [Author Response]

The following is the authors’ response to the original reviews.

We thank the reviewers for their insightful comments. The main issue raised by the reviewers was that because E6AP depletion reduced checkpoint signaling vis MASTL upregulation, this pathway is likely to be involved also in DNA damage checkpoint activation, in addition to checkpoint recovery. Hence, the proposed “timer”-like model is not fully supported. However, it is important to note that, the expression level of MASTL is not upregulated during the activation stage of the DNA damage checkpoint (unless E6AP is depleted). DNA damage signaling, via ATM-dependent E6AP phosphorylation, caused MASTL accumulation over time. This ultimately shifts the balance toward checkpoint recovery and cell cycle re-entry. As such, the role of MASTL (and E6AP-depletion) in suppressing DNA damage checkpoint is in harmony with the proposed role of MASTL upregulation in promoting checkpoint recovery. We have made additional clarifications about this point in the revised manuscript.

We have also addressed other concerns raised by the reviewers, as explained in the point-to-point responses below. With the addition of new modifications and data, we believe the revised manuscript is complete and conclusive.

**Reviewer #1 (Public Review):**
In principle a very interesting story, in which the authors attempt to shed light on an intriguing and poorly understood phenomenon; the link between damage repair and cell cycle re-entry once a cell has suffered from DNA damage. The issue is highly relevant to our understanding of how genome stability is maintained or compromised when our genome is damaged. The authors present the intriguing conclusion that this is based on a timer, implying that the outcome of a damaging insult is somewhat of a lottery; if a cell can fix the damage within the allocated time provided by the "timer" it will maintain stability, if not then stability is compromised. If this conclusion can be supported by solid data, the paper would make a very important contribution to the field.However, the story in its present form suffers from a number of major gaps that will need to be addressed before we can conclude that MASTL is the "timer" that is proposed here. The primary concern being that altered MASTL regulation seems to be doing much more than simply acting as a timer in control of recovery after DNA damage. There is data presented to suggest that MASTL directly controls checkpoint activation, which is very different from acting as a timer. The authors conclude on page 8 "E6AP promoted DNA damage checkpoint signaling by counteracting MASTL", but in the abstract the conclusion is "E6AP depletion promoted cell cycle recovery from the DNA damage checkpoint, in a MASTL-dependent manner". These 2 conclusions are definitely not in alignment. Do E6AP/MASTL control checkpoint signaling or do they control recovery, which is it?Also, there is data presented that suggest that MASTL does more than just controlling mitotic entry after DNA damage, while the conclusions of the paper are entirely based on the assumption that MASTL merely acts as a driver of mitotic entry, with E6AP in control of its levels. This issue will need to be resolved.

We thank the reviewer for his/her insightful comments. The main issue raised by the reviewers was that because E6AP depletion reduced checkpoint signaling vis MASTL upregulation, this pathway is likely to be involved also in DNA damage checkpoint activation, in addition to checkpoint recovery. Hence, the proposed “timer”-like model is not fully supported. However, it is important to note that, the expression level of MASTL is not upregulated during the activation stage of the DNA damage checkpoint (unless E6AP is depleted). DNA damage signaling, via ATM-dependent E6AP phosphorylation, caused MASTL accumulation over time. This ultimately shifts the balance toward checkpoint recovery and cell cycle re-entry. As such, the role of MASTL (and E6AP-depletion) in suppressing DNA damage checkpoint is in harmony with the proposed role of MASTL upregulation in promoting checkpoint recovery. We have made additional clarifications about this point in the revised manuscript.

As suggested by the reviewer, we have rephrased the statement in abstract to “E6AP depletion reduced DNA damage signaling, and promoted cell cycle recovery from the DNA damage checkpoint, in a MASTLdependent manner”.

As a mitotic kinase, MASTL promotes mitotic entry and progression. This is well in line with our findings that DNA damage-induced MASTL upregulation promotes cell cycle re-entry into mitosis. MASTL upregulation could also inhibit DNA damage signaling. This manner of feedback, inhibitory, modulation of DNA damage signaling by mitotic kinases (e.g., PLK1, CDK) has been implicated in previous studies (reviewed in Cell & Bioscience volume 3, Article number: 20 (2013)). In the revised manuscript, we have included more discussions about this aspect of checkpoint regulation.

Finally, the authors have shown some very compelling data on the phosphorylation of E6AP by ATM/ATR, and its role in the DNA damage response. But the time resolution of these effects in relation to arrest and recovery have not been addressed.

Detailed time point information is now added in the figure legends for E6AP phosphorylation data. We were able to observe this event during early stages (e.g., 1 hr, or 2-4 hr) of the DNA damage response, prior to significant MASTL protein accumulation.

**Reviewer #2 (Public Review):**
This is an interesting study from Admin Peng's laboratory that builds on previous work by the PI implicating Greatwall Kinase (the mammalian gene is called MASTL) in checkpoint recovery.The main claims of this study are:1. Greatwall stability is regulated by the E6-AP ubiquitin ligase and this is inhibited following DNA damage in an ATM dependent manner.1. Greatwall directly interacts with E6-AP and this interaction is suppressed by ATM dependent phosphorylation of E6-AP on S2181. E6-AP mediates Greatwall stability directly via ubiqitylation1. E6-AP knock out cells show reduced ATM/ATR activation and quicker checkpoint recovery following ETO and HU treatment1. Greatwall mediated checkpoint recovery via increased phosphorylation of Cdk substratesIn my opinion, there are several interesting findings presented here but the overall model for a role of the E6-AP -Greatwall axis is not fully supported by the current data and will require further work. Moreover, there are a number of technical issues making it difficult to assess and interpret the presented data.Major points:1. The notion that Greatwall is indeed required for checkpoint recovery hinges on two experiments shown in Figures 5A and B where Greatwall depletion blocks the accumulation of HELA cells in mitosis following recovery from ETO treatment and in G2/M following release from HU. An alternative possibility to the direct involvement of Greatwall in checkpoint recovery could be that Greatwall in HeLA cells is required for S-phase progression (as for example Charrasse et al. suggested). A simple control would be to monitor the accumulation of mitotic cells by microscopy or FACS following Greatwall depletion without any further checkpoint activation.

We thank the reviewer for his/her insightful comments.

Charrasse et al. showed ENSA knockout prolonged, but not stopped the progression of S-phase. In our experiments, MASTL (partial) knockdown did not significantly impact HeLa cells proliferation in the absence of DNA damage (Fig. 5, supplemental 1A). The reported role of MASTL in checkpoint recovery was consistently seen in response to various drugs, including etoposide which typically induces G2 arrest. Thus, we do not believe a prolonged S-phase accounts for the checkpoint recovery phenotype.

1. The changes in protein levels of Greatwall and the effects of E6-AP on Greatwall stability are rather subtle and depend mostly on a qualitative assessment of western blots. Where quantifications have been made (Figures 2D and 4F) the loading control and the starting conditions for Greatwall (0 timepoints in the right panel) appear saturated making precise quantification impossible. I would argue that the authors should at least quantify the immuno-blots that led them to conclude on changes in Greatwall levels and make sure that the exposure times used are in the dynamic range of the camera (or film). A more precise experiment would be to use the exogenously expressed CFP-Greatwall that is described in Figure 6 and measure the acute changes in protein levels using quantitative fluorescence microscopy in live cells. This is, in my opinion, a lot more trustworthy than quantitative immuno-blots.I also note here that most experiments linking Greatwall levels to E6-AP were done using siRNA, while the E6-AP ko cells would be a more reliable background for these experiments, especially with reconstituted controls.

DNA damage-induced MASTL upregulation was observed in various cell lines and after different treatments. To further strengthen this point, as suggested by the reviewer, we have included quantification of fluorescent measurements (Fig. 2, supplemental 1 A-C). Quantification of immunoblots for MASTL upregulation was also added in Fig. 1, supplemental 1E.The effects of E6AP depletion were consistently shown for both siRNA and stable KO.

1. This study has no data linking the effects of Greatwall to its canonical target PP2A:B55. The model shown in Figure 9 is therefore highly speculative. The possibility that Greatwall could act independently of PP2A:B55 should at least be considered in the discussion given the lack of experimental evidence.

The role of MASTL in promoting cell cycle progression via suppressing PP2A/B55 has been well established. As suggested by the reviewer, we have included discussions to acknowledge that “The role of MASTL upregulation in promoting checkpoint recovery and cell cycle progression can be attributed to inhibition of PP2A/B55, although the potential involvement of additional mechanisms is not excluded”.

1. The major effect of E6-AP depletion on the checkpoint appears to be a striking reduction in ATM/ATR activation, suggesting that this ubiquitin ligase is involved in checkpoint activation rather than recovery. It is not clear if this phenotype is dependent on Greatwall. If so it would be hard to reconcile with the default model that E6-AP acts via the destabilisation of Greatwall. In the permanent absence of E6-AP, increased Greatwall levels should inactivate B55:PP2A. How would this lead to a decrease in ATM/ATR activation? This is unlikely, and indeed Figure 5E shows that the reduction of MASTL in parallel to E6-AP does not result in elevated levels of ATR/ATM activation. Conversely, the S215A E6-AP mutant does have a strong rescue impact on ATR/ATM (Figure 8D).

We do not propose that PP2A/B55 directly dephosphorylates ATM/ATR-mediated phosphorylation. In fact, PP2A/B55 dephosphorylates and inactivates mitotic kinases and substrates which can feedback inhibit DNA damage checkpoint signaling (as previously shown for PLK1 and CDK). We included in a discussion about this point in the revised manuscript.

The point regarding checkpoint activation vs recovery is addressed below (point 5).

1. In summary, I do not think that the presented experiments clearly dissect the involvement of E6-AP and Greatwall in checkpoint activation and recovery. E6-AP depletion has a strong effect on checkpoint activation while Greatwall depletion is likely to have various checkpoint-independent effects on cell cycle progression.

It is important to note that, the expression level of MASTL is not upregulated during the activation stage of the DNA damage checkpoint (unless E6AP is depleted). DNA damage signaling, via ATM-dependent E6AP phosphorylation, caused MASTL accumulation over time. This ultimately shifts the balance toward checkpoint recovery and cell cycle re-entry. As such, the role of MASTL (and E6APdepletion) in suppressing DNA damage checkpoint is in harmony with the proposed role of MASTL upregulation in promoting checkpoint recovery. We have made additional clarifications about this point in the revised manuscript.

**Reviewer #3 (Public Review):**
In this manuscript, Li et al. describe the contribution of the ATM-E6AP-MASTL pathway in recovery from DNA damage. Different types of DNA damage trigger an increase in protein levels of mitotic kinase MASTL, also called Greatwall, caused by increased protein stability. The authors identify E3 ligase E6AP to regulate MASTL protein levels. Depletion or knockout of E6AP increases MASTL protein levels, whereas overexpression of E6AP leads to lower MASTL levels. E6AP and MASTL were suggested to interact in conditions without damage and this interaction is abrogated after DNA damage. E6AP was shown to be phosphorylated upon DNA damage on Ser218 and a phosphomimicking mutant does not interact with MASTL. Stabilization of MASTL was hypothesized to be important for recovery of the cell cycle/mitosis after DNA damage.The identification of this novel pathway involving ATM and E6AP in MASTL regulation in the DNA damage response is interesting. However, is surprising that authors state that not a lot is known about DNA damage recovery while Greatwall and MASTL have been described to be involved in DNA damage (checkpoint) recovery. In addition, PP2A, a phosphatase downstream of MASTL is a known mediator of checkpoint recovery, in addition to other proteins like Plk1 and Claspin. Although some of the publications regarding these known mediators of DNA damage recovery are mentioned, the discussion regarding the relationship to the data in this manuscript are very limited.

We thank the reviewer for his/her insightful comments. As suggested, the previously reported role of PLK1 and other cell cycle kinases in DNA damage checkpoint recovery is discussed in more details in the revised manuscript. As for PP2A/B55, we do not think it promotes checkpoint recovery, e.g., by dephosphorylating ATM/ATR or their substrates. Instead, this phosphatase dephosphorylates cell cycle kinases or their substrates, such as CDK1 or PLK1.

The regulation of MASTL stability by E6AP is novel, although the data regarding this regulation and the interaction are not entirely convincing. In addition, several experiments presented in this paper suggest that E6AP is (additionally) involved in checkpoint signalling/activation, whereas the activation of the G2 DNA damage checkpoint was described to be independent of MASTL. Has E6AP multiple functions in the DNA damage response or is ATM-E6AP-MASTL regulation not as straightforward as presented here?Altogether, in my opinion, not all conclusions of the manuscript are fully supported by the data.

We showed that E6AP depletion reduced checkpoint signaling vis MASTL upregulation, so this pathway is likely to be involved also in DNA damage checkpoint activation, in addition to checkpoint recovery. However, it is important to note that, the expression level of MASTL is not upregulated during the activation stage of the DNA damage checkpoint (unless E6AP is depleted). DNA damage signaling, via ATM-dependent E6AP phosphorylation, caused MASTL accumulation over time. This ultimately shifts the balance toward checkpoint recovery and cell cycle re-entry. As such, the role of MASTL (and E6APdepletion) in suppressing DNA damage checkpoint is in harmony with the proposed role of MASTL upregulation in promoting checkpoint recovery. We have made additional clarifications about this point in the revised manuscript.

**Reviewer #1 (Recommendations For The Authors):**
In principle a very interesting story, that attempts to shed light on an intriguing and poorly understood phenomenon; the link between damage repair and cell cycle re-entry once a cell has suffered from DNA damage. The issue is highly relevant to our understanding of how genome stability is maintained or compromised when our genome is damaged. The authors present the intriguing conclusion that this is based on a timer, implying that the outcome of a damaging insult is somewhat of a lottery; if a cell can fix the damage within the allocated time it will maintain stability, if not then stability is compromised. However, the story in its present form suffers from a number of major gaps that will need to be addressedMajor point:My primary concern regarding the main conclusion is that altered MASTL regulation seems to be doing much more than simply promoting more rapid recovery after DNA damage. This concern comes from the following gaps that I noted whilst reading the paper:Knock out of E6AP, is leading to a dramatic inhibition of ATM/ATR activation after damage (Fig.5C,D,E), this is (partially) rescued by co-depletion of MASTL (Fig5E). The authors will have to show that the primary effect of altered MASTL regulation is improved recovery, rather than reduced checkpoint activation. In other words, is initial checkpoint activation in cells that have lost E6AP normal, or do these cells fail to mount a proper checkpoint response? If the latter is true, that could completely alter the take home-message of this paper, because it could mean that E6AP/MASTL do not act as a "timer", but as a "tuner" to set checkpoint strength at the start of the DNA damage response. The authors themselves conclude on page 8 "E6AP promoted DNA damage checkpoint signaling by counteracting MASTL", but in the abstract the conclusion is "E6AP depletion promoted cell cycle recovery from the DNA damage checkpoint, in a MASTL-dependent manner". These 2 conclusions are definitely not in alignment, do E6AP/MASTL control checkpoint signaling or do they control recovery?

The expression level of MASTL is not upregulated during the activation stage of the DNA damage checkpoint (unless E6AP is depleted). DNA damage signaling, via ATM-dependent E6AP phosphorylation, caused MASTL accumulation over time. This ultimately shifts the balance toward checkpoint recovery and cell cycle re-entry. As such, the role of MASTL (and E6AP-depletion) in suppressing DNA damage checkpoint is in harmony with the proposed role of MASTL upregulation in promoting checkpoint recovery. We have made additional clarifications about this point in the revised manuscript. We have also made clarification to the statement indicated by the reviewer.

MASTL KD has a rather unexpected effect on cell cycle progression after HU synchronization (Fig.5B). It seems that the MASTL KD cells fail to exit from the HU-imposed G1/S arrest, an effect that is not rescued in the E6AP knock-outs. Inversely, E6AP knock-outs seem to more readily exit from the HU-imposed arrest, an effect that is completely lost after knock-down of MASTL. How do the authors interpret these results? Their conclusions are entirely based on a role for MASTL as a driver of mitotic entry, with E6AP in control of its levels, but this experiment suggests that MASTL and E6AP are controlling very different aspects of cell cycle control in their system.

As the reviewer pointed out, our data in checkpoint signaling and cell cycle progression suggested that MASTL upregulation could also inhibit DNA damage signaling, in addition to promoting cell cycle progression. This manner of feedback, inhibitory, modulation of DNA damage signaling by mitotic kinases (e.g., PLK1, CDK) has been implicated in previous studies (reviewed in Cell & Bioscience volume 3, Article number: 20 (2013)). In the revised manuscript, we have included discussions about this aspect of checkpoint regulation.

It is not possible to evaluate the validity of the conclusions that are based on Figure 6. We need to know how long the cells were treated with HU to disrupt the interaction between E6AP and MASTL. Is the timing of this in the range of the timing of MASTL increase after damage? A time course experiment is required here.The data obtained on E6AP-S218 phosphorylation and with the S218A mutant during damage and recovery look very promising. But again, the release from HU is confusing me as to what to conclude from them. Also, the authors should show how S218A expression affects MASTL levels (before and after damage). Also, a time course of ATM/ATR activation is required to decide if initial or late ATM/ATR signaling is affected.

Detailed time point information is now added in the figure legends for E6AP phosphorylation and E6AP-MASTL dissociation data. We were able to observe these events during early stages (e.g., 1 hr, or 2-4 hr) of the DNA damage response, prior to significant MASTL protein accumulation.

The conclusion that "and was not likely to be caused by the completion of DNA repair, as judged by the phosphorylation of replication protein A" (page 5) is based on western blots that represent the average across the entire population. It is possible that MASTL expression is still low in the cells that have not completed repair, while it's increase on blots comes from a subset of cells where repair is complete. The authors should perform immunofluorescence so that expression levels of MASTL can be directly compared to levels of phospho-RPA in individual cells. In fact, the manuscript could benefit a lot from a more in-depth single-cell (microscopy)-based analysis of the relations over time between ATM/ATR activation, E6AP phosphorylation, MASTL stabilization versus the checkpoint arrest and subsequent recovery.

Time point analyses were provided for DNA damage-induced RPA phosphorylation and ATM/ATR substrate phosphorylation (Fig. 1). These data showed MASTL accumulation in the presence of active DNA damage checkpoint signaling. To further strengthen this point, as suggested by the reviewer, we have included quantification of fluorescent measurements (Fig. 2, supplemental 1 A-C). IF data showed MASTL upregulation in correlation with ATM/ATR activation.

Minor points:It's not "ionized radiation", but "ionizing radiation" (page 5)

We have made the correction as pointed out by the reviewer.

Expression levels of MASTL should be quantified over time after DNA damage. In some of the experiments the increase seems to plateau relatively quick (HU treatment, fig 1B, 1-2 hours), while in others the levels continue to increase over longer periods (HU treatment, fig 1D, 6 hours). This is relevant to the timer function of MASTL that is proposed here.

The kinetics of MASTL upregulation is generally consistent among all cell lines. As suggested, quantification of immunoblots is provided (Fig. 1, supplemental 1E); additional quantification of IF signals is also included (Fig. 2, supplemental 1 A-C).

The experiment executed with caffeine (page 5) should be repeated with more selective/potent ATM/ATR inhibitors that are commercially available.

Specific ATM inhibitor was used to confirm the caffeine result in Fig. 7 supplemental 1B&C.

"a potential binding pattern" (page 6) should be "a potential binding partner"

We have made the correction as pointed out by the reviewer.

**Reviewer #2 (Recommendations For The Authors):**
1. All western blots require size markers. The FACS blots shown do not have any axis labels.

We have included size markers for blots, at the first appearance of each antibody. Labels are added for FACS blots.

1. The quantification of mitotic cells does not indicate how many cells were counted and if this was done by eye or using software.

The missing experimental information is included in the figure legends, as suggested.

1. The western blots demonstrating ubiquitylation of Greatwall (Figure 4D) are of very poor quality and impossible to interpret.

The ubiquitination of MASTL did not show clear ladders, possibly due to its relative protein size.

**Reviewer #3 (Recommendations For The Authors):**
Specific suggestions to improve the manuscript:1. Include literature regarding known mediators of DNA damage checkpoint recovery, including MASTL/Greatwall and PP2A, in the manuscript and discuss the observations from this manuscript in relationship with the literature.

Related literatures are included in the discussion.

1. The increase in MASTL protein levels upon DNA damage are not always clear, for example Fig. 1A. The same for MASTL stability after DNA damage, such as in Fig. 2C. Quantification of the westerns would help demonstrating a significant effect.

As suggested by the reviewer, we have included quantification of fluorescent measurements (Fig. 2, supplemental 1 A-C). Quantification of immunoblots for MASTL upregulation was also added in Fig. 1, supplemental 1E.

1. The E6AP-MASTL in vitro interaction studies shown in Fig. 3 raise doubts. First, beads only are used as negative control, whereas MBP only-beads are a better control. The westerns in top panels of 3B (MASTL), 3C (GST-MASTL) and 3D (MASTL) should be improved. In addition, in Fig. 3C, different GSTMASTL fragments are used in an MBP-E6AP pull down, but the GST-MASTL input does not show any specific band to demonstrate that these fragments are correct. The same for the GFP-E6AP fragments in Fig. 3 Suppl. 1C The input does not show any proteins, there is no N fragment present in the IP and the size of the fragment N3 in the IP GFP does not seem correct.Altogether, it makes me doubt that the interaction between E6AP and MASTL is direct. Better data with appropriate controls should show whether the interaction is direct or mediated via another protein.

Purified proteins used for the in vitro interaction had significant degradation, causing many bands in the input. We included a lighter exposure of the input here as Author response image 1. MBP alone did not bind MASTL, as both M and C segments of MASTL were MBP-tagged, and did not pull down MASTL. We agree with the reviewer that our direct interaction data showed rather weak MASTL/E6AP interaction, suggesting the interaction is dynamic or possibly mediated by additional binding proteins. We have included this statement in the revised manuscript “Taken together, our data characterized MASTL-E6AP association which was likely mediated via direct protein interaction, although the potential involvement of additional binding partners was not excluded”.

1. Fig. 4B. Overexpression of HA-E6AP results in a decrease in MASTL protein levels. Can this effect be rescued by treatment with proteasome inhibitor MG132?

As expected, MG132 stabilized MASTL, with or without E6AP overexpression. We have added this new data in Fig. 4, supplemental 1B.

1. Fig. 4G. MASTL interacts with HA-ubiquitin in WT, but not E6AP KO cells. These cells are treated with MG132, so if E6AP really ubiquitinates MASTL, I would expect MASTL to be polyubiquitinated. However, the "interaction signal" does not show polyubiquitination. In fact, this band actually runs lower than MASTL in input samples, which even could be an artifact. Please explain.

The ubiquitination of MASTL did not show clear ladders, possibly due to its relative protein size. As the reviewer noted, the band position in the HA-Ub IP lanes seemed slightly shifted, compared to the input. We have noticed in many experiments that bands in the IP lanes did not perfectly align with the input lanes.

1. The DNA damage recovery experiments measuring mitotic index after washing off etoposide (Fig. 5A and Fig. 8A): What are the time points taken? And importantly, why are there no error bars on these intermediate time points, but only on the 4 hour time point?

As suggested, time point information and additional error bars are included.

1. Fig. 5E. According to the authors, depletion of MASTL rescues the effect of KO of E6AP. However, no increase in pATM/ATR substrate signal is seen upon etoposide treatment in these samples so I am not convinced this experiment demonstrates a rescue.

The rescue was evident, especially for many high molecular weight bands which were more effectively detected by this phospho-specific antibody.

1. Fig. 5C and 8D strongly suggest that E6AP is involved in checkpoint activation. How do these data relate to DNA damage recovery? Is the recovery in E6AP KO cells faster as a consequence of reduced checkpoint signaling or is the recovery effect really specific by stabilization of MASTL? These data should be explained, also taken the data from Wong et al. (Sci. Rep. 2016) into account, that demonstrate that G2 checkpoint activation is independent of MASTL.

The expression level of MASTL is not upregulated during the activation stage of the DNA damage checkpoint (unless E6AP is depleted). DNA damage signaling, via ATM-dependent E6AP phosphorylation, caused MASTL accumulation over time. This ultimately shifts the balance toward checkpoint recovery and cell cycle re-entry. As such, the role of MASTL (and E6AP-depletion) in suppressing DNA damage checkpoint is in harmony with the proposed role of MASTL upregulation in promoting checkpoint recovery. We have made additional clarifications about this point in the revised manuscript.

1. The model presented in Fig. 9 is puzzling because there does not seem to be a difference between phosphorylation of E6AP and the interaction with MASTL on early versus late times after DNA damage. And this exactly is what is missing in the manuscript: A more detailed evaluation of the timing of E6APSer218 phosphorylation and the E6AP-MASTL interaction in response to DNA damage.

More clarification is given to explain this model in the figure legend of Fig. 9.

Time point analyses were provided for DNA damage-induced RPA phosphorylation and ATM/ATR substrate phosphorylation (Fig. 1). These data showed MASTL accumulation in the presence of active DNA damage checkpoint signaling. To further strengthen this point, we have included quantification of fluorescent measurements (Fig. 2, supplemental 1 A-C). IF data showed MASTL upregulation in correlation with ATM/ATR activation. Time point information was also added for Ser-218 phosphorylation and MASTL-ENSA dissociation which were observed in early stages of the DNA damage response (1 hr, or 2-4 hr).